analytical chemistry

buckwheat honey, botanical origin, polyphenol profile, flavan-3-ols, UHPLC MS orbitrap

**Author for correspondence:**
Živoslav Tešić
e-mail: ztesic@chem.bg.ac.rs

This article has been edited by the Royal Society of Chemistry, including the commissioning, peer review process and editorial aspects up to the point of acceptance.

# Polyphenol profile of buckwheat honey, nectar and pollen

Milica Nešović[1], Uroš Gašić[5], Tomislav Tosti[2],
Nikola Horvacki[3], Branko Šikoparija[6], Nebojša Nedić[7],
Stevan Blagojević[1], Ljubiša Ignjatović[4]
and Živoslav Tešić[2]

[1]Institute of General and Physical Chemistry, [2]Faculty of Chemistry, [3]Innovation Centre–Faculty of Chemistry, and [4]Faculty of Physical Chemistry, University of Belgrade, Studentski trg 12-16, Belgrade 11158, Serbia
[5]Department of Plant Physiology, Institute for Biological Research 'Siniša Stanković', National Institute of Republic of Serbia, University of Belgrade, Bulevar despota Stefana 142, Belgrade 11060, Serbia
[6]BioSense Institute - Research Institute for Information Technologies in Biosystems, University of Novi Sad, Novi Sad 21101, Serbia
[7]Faculty of Agriculture, Institute for Zootehnics, University of Belgrade, Nemanjina 6, Belgrade, Zemun 11080, Serbia

ŽT, 0000-0002-5162-3123

A focus of research in recent years is the comparison of honey as the final product of bees with pollen and nectar of the plant from which the honey originates, as the main food source for bees. Buckwheat honey is recognized as a nutritionally valuable product, which provides a scientifically proven health benefit and is confirmed as a functional food. The quality of this type of honey is attributed to high levels of phytochemicals in buckwheat. The purpose of this study was the examination of similarity between buckwheat honey and buckwheat nectar and pollen, as well as simultaneous investigation of their chemical profiles and the origin of the honey. The phenolic profile of buckwheat pollen showed a lower number of flavonoids and phenolic acids than those of nectar and honey samples, but confirmed the presence of the most characteristic polyphenols derived from the buckwheat plant. The notable difference was found to be the presence of (epi)catechin units, its galloylated derivatives and procyanidin dimers, which were not present in honey. Honey polyphenols displayed a pronounced correlation with those of nectar, but not with those of pollen. Finally, by comparing the polyphenolic profiles of honey, nectar and pollen sharing the same geographical origin, new data could be provided for a potential assessment of the botanical origin of buckwheat honey.

# 1. Introduction

Buckwheat is a pseudo-cereal from the Polygonaceae family, and as a multipurpose plant, it is recognized as an essential functional food [1]. Common buckwheat (*Fagopyrum esculentum* Moench) is a reliable and high-yielding honey plant [2]. A large number of flowers that bloom for a long time contribute to the buckwheat fields being a valuable and suitable bee pasture.

Buckwheat honey is highly valued, with specific organoleptic properties, such as strong flavour, characteristic colour, pungent taste and aroma of molasses, but not preferred by consumers [3]. Regardless of estimation by consumers, buckwheat honey has a very high nutritional value with beneficial antioxidant [4,5] and anti-inflammatory effects [6]. As compared with other types of honey, buckwheat honey showed a higher content of minerals, sugars and phenolic compounds [7–9].

Buckwheat is a good source of nutrients and bioactive compounds [1,2] which are significantly responsible for the antioxidant activity it possesses. High content of rutin was noted in all parts of buckwheat [10–13]. Monomeric flavan-3-ols (catechin and epicatechin) and procyanidins were found in buckwheat extracts [12–17] and proposed as buckwheat markers. As concerns the buckwheat honey *fingerprint*, many authors suggested the dominant presence of *p*-coumaric acid and *p*-hydroxybenzoic acid [4,5]. Monofloral honey should predominantly contain pollen from plant species that has declared that honey type. According to Beckh *et al.* [18], buckwheat honey must contain over 30% of buckwheat pollen. Such buckwheat monofloral honey is relatively rare. It is mostly found in countries with high rates of buckwheat production, such as China, Russia, Ukraine, as well as Poland, while in Serbia buckwheat fields are in a significantly smaller area [19]. Moreover, published data show high variability of *Fagopyrum* pollen frequency [3–5,8,9,20,21], whereas only Polish Standard [20] still define buckwheat honey as the one with over 45% of pollen. Variability in *Fagopyrum* pollen contribution in monofloral buckwheat honey [4], indicating a variability in nectar contribution [22], requires the detection of additional phytochemicals that could be useful for establishing the accurate classification of this type of honey.

The objective of this study was to characterize buckwheat honey samples from Serbia and Poland based on analysis of phenolic compounds, as well as on antioxidant tests (total phenolic content (TPC) and radical scavenging activity (RSA)), pollen analysis and physico-chemical parameters (moisture content, electrical conductivity and sugar profile). In addition to the generally accepted pollen analysis for determining the botanical origin of honey, analysis of polyphenols could be a method of choice, especially for honey with a low content of pollen. In order to provide the detection of specific phenolic compounds in buckwheat honey, high-resolution mass spectrometry (HRMS) in combination with multi stage mass spectrometry (MS$^n$) was used. To obtain a more accurate correlation of phenolic compounds with the botanical origin of honey, the identification of phenolic compounds in buckwheat nectar and buckwheat pollen samples was also performed. By correlating the polyphenolic profile of buckwheat honey, nectar and pollen, we wished to obtain additional important data for the assessment of the botanical origin of honey with low pollen content. For this purpose, four samples of honey from Serbia with a low content of buckwheat pollen particles declared by beekeepers as buckwheat honey, and two buckwheat honey samples from Poland with a high content of buckwheat pollen were analysed. With this in mind, two carefully selected samples of honey, containing the low and high percentage of buckwheat pollen, were subjected to a more detailed study of polyphenol profiles.

# 2. Material and methods

## 2.1. Samples

For this study, six buckwheat honey samples (collected by bees *Apis mellifera*) were analysed (table 1). Four honey samples (assigned as H1-H4) were provided by Serbian beekeepers, during 2017, from hives situated within a small geographical area (of about 2500 km$^2$) in western Serbia (figure 1). Buckwheat honey samples H5 and H6 (from two separate locations) were received from the commercial market in southern Poland in 2016 (figure 1). Buckwheat nectar and pollen were collected directly from flowers of buckwheat cultivated in the village of Radijevići (figure 1). They were harvested in July 2017, during the nectar secretion of flowers. The pollen was sampled by scattering the buckwheat flowers, while nectar was sucked out of the flowers using micro-capillaries, and kept in Eppendorf test tubes. Samples were stored in the refrigerator until the analysis.

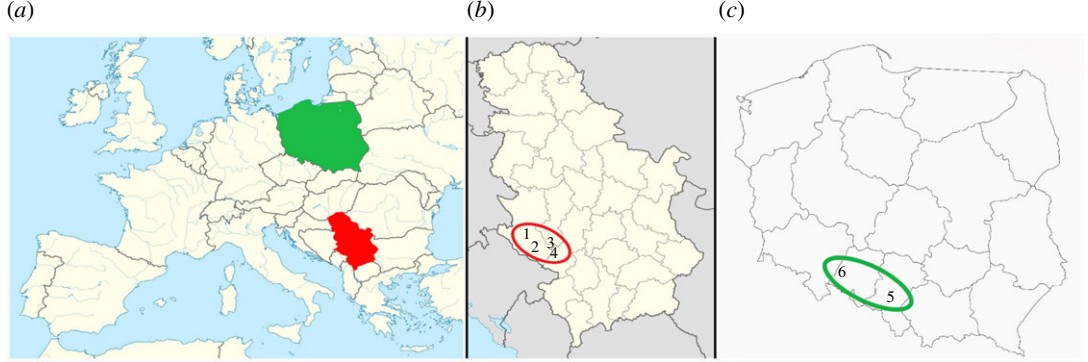

**Figure 1.** (*a*) Regional map of Serbia (red) and Poland (green). (*b*) Map of sampling sites in Serbia: 1, Radijeviċi, Nova Varoš (43°23′31″ N; 19°52′20″ E); 2, Sjenica (43°16′14″ N; 19°59′35″ E); 3, Teočin, Ravna Gora (44°04′03″ N; 20°14′02″ E); 4, Teočin, Ravna Gora (44°03′44″ N; 20°14′42″ E). (*c*) Map of sampling sites in Poland: 5, Katowice (50°16′00″ N; 19°01′00″ E); 6, Starowice (50°38′36″ N; 17°19′40″ E).

**Table 1.** List of investigated honey samples and melissopalynological analysis, which declared samples to be buckwheat honey types after pollen from nectarless plants is excluded. (Pollen frequency classes: P, 'predominant pollen' (more than 45% of pollen grains counted), S, 'secondary pollen' (16–45%); I, 'important minor pollen' (3–15%); M, 'minor important pollen' (less than 3%).)

| sample | location | % of *Fagopyrum* pollen | secondary pollen (16–45%) | important minor pollen (3–15%) |
|---|---|---|---|---|
| H1 | Serbia | 4.05 | *Echium*, *Astragalus* type | *Fagopyrum*, *Achilea* type, *Aster* type, *Filipendula* |
| H2 | Serbia | 8.91 | *Astragalus* type | *Fagopyrum*, *Filipendula*, *Lamiaceae* S type, *Hypericum*, *Brassicaceae*, *Amorpha* type, |
| H3 | Serbia | 10.93 | *Filipendula*, *Amorpha* type | *Fagopyrum*, *Astragalus* type, *Rubus*, *Teucrium*, *Clematis* |
| H4 | Serbia | 4.47 | *Filipendula* | *Fagopyrum*, *Rosaceae* (*Prunus* type), *Rubus*, *Astragalus* type, *Trifolium pratense*, *Fenestrate*, *Amorpha* type, *Tilia*, *Robinia*, *Teucrium*, *Rhamnus* type, |
| H5 | Poland | 40.83 | *Fagopyrum*, *Brassica napus* | *Trifolium pratense*, *Phacelia*, *Centaurea cyanus*, *Senecio* type |
| H6 | Poland | 24.29 | *Fagopyrum*, *Rubus*, *Brassicaceae* | *Astragalus* type, *Fenestrate* |

## 2.2. Chemical and materials

Folin-Ciocalteùs (FC) reagent, gallic acid, sodium carbonate, 2,2-diphenyl-1-picrylhydrazyl (DPPH), Trolox standard, sodium hydroxide, sodium acetate and all chemicals for chromatographic analysis of polyphenols (hydrochloric acid, acetonitrile, formic acid, and phenolic standards) were purchased from Sigma Aldrich (Steinheim, Germany). Sugar standards (glucose, fructose, sucrose, maltose, isomaltose, trehalose, turanose, melibiose and melezitose) were supplied from Tokyo Chemical Industry, Europe (Zwijndrecht, Belgium). Ultrapure water (18 MΩcm) produced by TKA MicroPure water purification system (Thermo Fisher TKA, Niederelbert, Germany) was used for the preparation of all aqueous solutions. Syringe filters (25 mm, polytetrafluoroethylene membrane 0.45 μm) supplied by Supelco (Bellefonte, PA, USA) was used for filtration of samples. Cartridges (Strata C18-E, 500 mg 3 ml$^{-1}$) for solid-phase extraction (SPE) were obtained from Phenomenex (Torrance, CA, USA).

## 2.3. Melissopalynology analysis

Frequency of pollen was determined for all six honey samples. Pollen was extracted and analysed following the Harmonized Methods of Melissopalynology [23]. Microscopic slides were scanned until a minimum of 500 pollen grains was counted and identified using referent slides and pollen identification atlases [24]. Other elements i.e. algae, fungal spores and hyphae, anemophilous pollen and pollen of nectarless plants [25] were also counted. Relative frequency of identified pollen was calculated [26], and the frequency classes [27] were reported, including the values after nectarless species were excluded from the analysis.

## 2.4. Determination of physico-chemical parameters

According to the method described by Bogdanov *et al*. [28], moisture content, electrical conductivity and sugars content were determined. Content of sugars was analysed using high-performance anion-exchange chromatography with a pulsed amperometric detector (Dionex ICS 3000) with a previously described procedure for preparation of sugar standards [29]. The refractive index of honey samples was measured using an Abbe-type refractometer (Atago RX refractometer, Japan) at a temperature of 20°C. After using the Chataway's table correction on these values, the results of the moisture content of honey samples were obtained. Conductivity metre (WTW Golden Laboratory & Engineering, Cond 7110, Inolab) with a conductivity cell with its constant of $0.108 \, cm^{-1}$ was used for measuring the electrical conductivity of 20% (w/v) aqueous solution of honey.

## 2.5. Phenolic profile

The preparation of samples and standard solutions for phenolic profile analysis was achieved by following the previously described procedure by Gašić *et al*. [29]. Stock solutions of available phenolic standards were prepared in methanol in a concentration range of 0.025 to $2.000 \, mg \, l^{-1}$. Isolation of phenolic compounds from samples was carried out using SPE.

Qualitative analysis of phenolic compounds was carried out by an Accela 600 ultra-high performance liquid chromatography (UHPLC) system connected to an LTQ OrbiTrap mass spectrometer (Thermo Fisher Scientific, Bremen, Germany). Chromatographic and spectrometric conditions, as well as instruction for fragmentation study, were previously described by Gašić *et al*. [29]. The presence of some phenolic compounds was confirmed using available standards, while the other was identified by examination of previously published $MS^2$, $MS^3$ or $MS^4$ fragmentation data. Tentative identification of some phenolic compounds in the absence of standards was achieved by HRMS and $MS^4$ fragmentation.

Quantitative analysis of phenolic compounds was performed using a Dionex ICS 3000 UHPLC system connected to a TSQ Quantum Access Max triple-quadrupole mass spectrometer (UHPLC-QqQ-MS/MS, Thermo Fisher Scientific, Basel, Switzerland) followed by chromatographic and mass spectrometry conditions such as those previously used by Gašić *et al*. [30]. Quantification of phenolic compounds was based on a direct comparison with available standards, and results were expressed as $mg \, kg^{-1}$.

## 2.6. Antioxidant tests

Spectrophotometric determinations of TPC and RSA were performed in the reaction with FC reagent and DPPH, respectively, by following the procedures previously described by Nešović *et al*. [31]. A UV/Vis spectrophotometer (GBC UV/Visible Cintra 6, Australia) was used for the recording of wavelengths.

# 3. Results and discussion

## 3.1. Melissopalynology analysis

The dominant presence of pollen grains from nectarless plant species (e.g. *Fraxinus americana*) was noticed in analysed buckwheat honey samples (electronic supplementary material, table S1). This ornamental species that flowers before buckwheat during April and May can 'contaminate' nectar stored in the comb. As the bees collect pollen to obtain many proteins, minerals, and lipids they cannot find in nectar [32], they often visit different types of plants to collect pollen with a suitable

composition. Therefore, the pollen grains present in honey do not necessarily originate from the same species as nectar, as also pointed out by others [33].

After the removal of pollen from nectarless species, the qualitative melissopalynological analysis identified *Fagopyrum* pollen at the levels of 4.05–10.93% in the analysed Serbian buckwheat honey, compared to 40.83% and 24.29% in the Polish honey assigned as H5 and H6, respectively. *Fagopyrum* pollen was important minor pollen in Serbian (3–15%) and the secondary pollen (16–45%) in the Polish honey samples declared as buckwheat honey (table 1).

Beside *Fagopyrum*, pollen from nectariferous plants belonging to Boraginaceae, Rosaceae, Fabaceae, Brassicaceae, Asteraceae, Hypericaceae, Violaceae and Malvaceae families, was identified in both Serbian and Polish honey samples. None of the identified pollen can be classified as predominant. Pollen grains from *Astragalus* and *Amorpha* types (Fabaceae), *Echium* (Boraginaceae) and *Filipendula* (Rosaceae) were found as the secondary pollen in Serbian samples. Buckwheat honey from China also contained a high content of pollen grains from *Astragalus* [34]. In analysed Polish buckwheat honey samples, *Fagopyrum* was the secondary pollen together with *Rubus*, *Brassica napus* and other Brassicaceae.

A wide range of pollen grains was identified in the Polish buckwheat honey and the presence of significant amounts of *B. napus* and *Phacelia* pollen, as well as some *Helianthus* pollen indicates that the bees were foraging in a more agricultural area that it is the case for the Serbian samples. The other pollen types recorded in Polish honey samples correspond to the spectrum discovered by others [20]. However, the quantity of that pollen in our samples could not influence its classification.

As it appears from the results of the melissopalynological analysis, our honey samples declared as buckwheat have a widespread range of *Fagopyrum* pollen. In view of the fact, there are different types of monofloral buckwheat honey with respect to the contribution of *Fagopyrum* pollen [4]; the Polish honey samples have an appropriate declaration, while the Serbian honey samples could be only potentially considered as buckwheat honey. These results partially confirmed observations of other authors who reported high variability of *Fagopyrum* pollen in buckwheat honey, from 0.3 to 86% [3–5,8,9,20,21,35].

Generally, buckwheat pollen is not expected to be under-represented in honey [22]. The size of *Fa. esculentum* Moench pollen ranges from 26 to 50 µm indicating it could be more easily filtered out during nectar transport especially if sources are distant from the hive. However, high variability in the contribution of *Fagopyrum* pollen in honey could be owing to the fact there are two types of flowers on buckwheat, which do not have the same flow of pollen and nectar [36]. Different pollen flow for distylous species was already noted for the plants from Rubiaceae family [37]. Bees collect pollen to obtain many proteins, minerals and lipids they cannot find in nectar [32]. Therefore, during *Fagopyrum* nectar flow, bees are forced to visit various sources to satisfy their feeding requirements. In addition, bees also pick up pollen by touching stamens of several flowers. Consequently, the pollen grains present in honey do not necessarily originate from the same species as nectar [33]. As a result, honey can occur with under-representative pollen, meaning that the frequency of pollen in honey does not correlate with the corresponding nectar contribution [25]. Furthermore, in countries with low buckwheat production, such as Serbia, there are small areas with cultivated buckwheat, making the source of food for bees limited.

## 3.2. Physico-chemical analysis

Comparing the results of physico-chemical parameters for the analysed Serbian and Polish buckwheat honey samples (table 2), there were no pronounced differences. The obtained values of moisture content are below 17.1% and in accordance with the regulations for honey [38], the same content of monosaccharides (higher than 60 g per 100 g) and sucrose (less than 3 g per 100 g). The values for electrical conductivity of the Serbian honey were between 0.32 and 0.43 mS cm$^{-1}$ which is very similar to published data [35,39], while for the Polish honey a trace lower, 0.21 mS cm$^{-1}$ and 0.28 mS cm$^{-1}$. Other published values were higher, such as for buckwheat honey from Poland, i.e. in the range of 0.40–0.53 mS cm$^{-1}$ [20], as well as for buckwheat honey from China, i.e. 0.35–0.63 mS cm$^{-1}$ [5].

The sugar profile of buckwheat honey, besides dominant fructose and glucose, contains oligosaccharides typical for blossom honey, such as turanose, maltose and trehalose [40]. Determined values of total sugars (from 67.00 to 71.53 g per 100 g) were lower than other authors reported [4,41]. Melezitose was found in our six analysed buckwheat honey samples, while others [4,41] did not find it.

Our results of physico-chemical parameters confirmed observations of Popek *et al.* [39], who proposed distinguishing of buckwheat honey based on a Decision Tree developed classification model

**6**

**Table 2.** Physico-chemical parameters (water content, electrical conductivity, sugar content) of buckwheat honey samples from Serbia (H1–H4) and Poland (H5, H6). (Mean ± s.d., mean value ± standard deviation (p ≤ 0.05).)

| physico-chemical parameters | Serbian honeys | | | | | Polish honeys | | |
|---|---|---|---|---|---|---|---|---|
| | H1 | H2 | H3 | H4 | mean ± s.d. | H5 | H6 | mean ± s.d. |
| moisture content (%) | 17.10 | 17.03 | 15.55 | 15.50 | 16.29 ± 0.89 | 15.50 | 15.55 | 15.53 ± 0.04 |
| electrical conductivity (mS per cm$^{-1}$) | 0.41 | 0.43 | 0.35 | 0.32 | 0.38 ± 0.05 | 0.279 | 0.212 | 0.25 ± 0.05 |
| sugars (g per 100 g) | 67.00 | 69.73 | 68.46 | 67.39 | 68.15 ± 1.06 | 71.53 | 71.16 | 71.34 ± 0.26 |
| glucose (g per 100 g) | 25.91 | 28.47 | 25.12 | 24.57 | 26.02 ± 1.72 | 27.51 | 26.70 | 27.11 ± 0.57 |
| fructose (g per 100 g) | 37.39 | 37.26 | 39.03 | 38.61 | 38.07 ± 0.88 | 39.61 | 39.39 | 39.50 ± 0.16 |
| sucrose (g per 100 g) | 1.304 | 1.301 | 1.300 | 1.279 | 1.30 ± 0.01 | 1.286 | 1.789 | 1.54 ± 0.36 |
| maltose (g per 100 g) | 0.681 | 0.633 | 0.537 | 0.532 | 0.60 ± 0.07 | 0.676 | 0.741 | 0.71 ± 0.05 |
| isomaltose (g per 100 g) | 0.418 | 0.593 | 0.804 | 0.773 | 0.65 ± 0.18 | 0.695 | 0.606 | 0.65 ± 0.06 |
| trehalose (g per 100 g) | 0.123 | 0.151 | 0.186 | 0.182 | 0.16 ± 0.03 | 0.078 | 0.079 | 0.08 ± 0.00 |
| turanose (g per 100 g) | 0.721 | 0.908 | 0.998 | 0.948 | 0.89 ± 0.12 | 0.926 | 0.958 | 0.94 ± 0.02 |
| melibiose (g per 100 g) | 0.252 | 0.213 | 0.264 | 0.247 | 0.24 ± 0.02 | 0.353 | 0.359 | 0.36 ± 0.00 |
| melezitose (g per 100 g) | 0.209 | 0.207 | 0.221 | 0.252 | 0.22 ± 0.02 | 0.388 | 0.530 | 0.46 ± 0.10 |
| sum of monosaccharides (g per 100 g) | 63.29 | 65.72 | 64.16 | 63.18 | 64.09 ± 1.17 | 67.13 | 66.09 | 66.61 ± 0.73 |
| sum of disaccharides (g per 100 g) | 3.50 | 3.80 | 4.09 | 3.96 | 3.84 ± 0.25 | 4.01 | 4.53 | 4.27 ± 0.37 |

with the following parameters: electrical conductivity ≤0.45 mS cm$^{-1}$, sucrose ≤3.105 g per 100 g and reducing sugars ≤78.525 g per 100 g.

## 3.3. Phenolic profile

Comparing mass chromatograms with available standards or in the absence of the standards, with published MS fragmentation pathways [12–15,17,29], we identified 55 phenolic compounds. Results of the qualitative analysis of two buckwheat honeys with low and high content of pollen particles (H1 and H5), as well as buckwheat nectar (N) and pollen (P), are presented in table 3. According to the number of identified phenolic compounds, the analysed buckwheat samples may be arranged as follows: Polish buckwheat honey (41) > Serbian buckwheat honey (40) > buckwheat nectar (33) > buckwheat pollen (21). The most common compounds, according to the number identified, were flavonoids, followed by phenolic acids and flavonoid glycosides (table 3).

Phenolic acids were identified as free acids and in the form of hexosides and esters. Two derivatives of caffeoylquinic acid (compounds 4 and 6) were found in honey samples, while 5-$O$ derivative (chlorogenic acid) was found only in nectar and pollen. Two more quinic acid esters were found in the tested samples; compound 9 (5-$O$-$p$-coumaroylquinic acid) only in buckwheat honey from Serbia and compound 10 (methyl 5-$O$-caffeoylquinate) only in the pollen sample. Identification of all these quinic acid derivatives, including esterification position, was in accordance with previously published MS and chromatographic data [42].

Propolis-derived flavonoids (chrysin, pinocembrin, pinobanksin and galangin) were found in both honey and buckwheat nectar samples. As nectar is the main source of food for bees, it is natural for the phenolic profile to be similar to honey. The phenolic compounds present in honey came from buckwheat nectar but also from many other plant species that bees are visiting, which resulted in the presence of more other phenolic compounds in honey. Compound 42 was not identified only in the pollen sample, so it can be said to originate from nectar or propolis. Glycoside derivatives of this compound have previously been found in *B. napus* honey [43], while the detailed fragmentation pathway of compound 42 was shown in our previous study [44]. Likewise, pinobanskin derivatives (compounds 49, 51–54) were not found only in the tested pollen sample, but they are known to be propolis-derived compounds and their fragmentation is well described in the literature [45]. Phenolic compounds were recorded that differentiate buckwheat nectar from corresponding honey, but they did not include flavonoid glycosides (table 3). However, flavonoid glycosides are considered to dominate in honey because of hydrolysis in the presence of bee-enzymes [4]. Nevertheless, three flavonol 3-$O$-glycosides (quercetin 3-$O$-rhamnoside, quercetin 3-$O$-(6″-rhamnosyl)-glucoside (rutin) and kaempferol 3-$O$-rhamnoside) were common for nectar and Serbian honey, as well as pollen. Quercetin 3-$O$-galactoside (hyperoside) was not identified only in the buckwheat nectar sample. Glycosylation position of all identified flavonol glycosides (compounds 35–38) were confirmed by the presence of the MS$^2$ base peak at mass of aglycone and high intensity of its radical ion ($m/z$ 300 for quercetin and $m/z$ 284 for kaempferol) [46]. Generally, rutin also appears in different parts of buckwheat [12–16]. The presence of kaempferol rhamnosides was identified in both nectar and honey samples from *Robinia pseudacacia* [47]. Later, the presence was reported of kaempferol rhamnosides and rhamnosyl-glucosides, through pollen, nectar and honey from *Diplotaxis tenuifolia* [48].

Aglycones are more potent antioxidants than corresponding glycosides, but the glucose portion can enhance the bioavailability of compounds [49]. Isomers of flavone 6-$C$-glycosides and flavone 8-$C$-glycosides were differentiated in buckwheat sprouts [50]. In our study, luteolin 6-$C$-hexoside and luteolin 8-$C$-hexoside were found in buckwheat honey from Poland.

As to the variety of the identified phenolic compounds in the analysed buckwheat samples, a smaller number of phenolic acids, as well as flavonoids, was found in pollen, while the group of $O$-methylated flavonoids generally was not present in the pollen. Of 21 phenolic compounds found in the pollen, only myricetin ($m/z$ 317), which is a pollen-nectar-derived flavonoid [51], was common to pollen and honey from Serbia. The presence of myricetin in buckwheat honey has been reported [35,45] and also in buckwheat flours (in a bound form) [15]. Furthermore, myricetin has been proposed as a chemical marker of the authenticity of heather honey [7].

Phenolic compounds that were present only in the pollen sample were mostly (epi)catechin units, its galloylated derivatives and procyanidin dimers, which have already been considered as buckwheat markers [12–17,29]. Potential health effects have been reported of procyanidins B type with their content being higher in buckwheat than in cereals (barley and spelt) [17]. Additionally, procyanidins increase antioxidant effect against superoxide anions more than monomeric flavonoids [49].

**Table 3.** High-resolution MS data and negative ion MS², MS³ and MS⁴ fragmentation of phenolic compounds identified in Serbian buckwheat honey H1, Poland buckwheat honey sample H5, nectar (N) and pollen (P). (ND, not detected.)

| no. | compound name | $t_R$ (min) | molecular formula, [M−H]− | calculated mass, [M−H]− | exact mass, [M−H]− | Δ ppm | MS² fragments (% base peak) | MS³ fragments (% base peak) | MS⁴ fragments (% base peak) | H1 | H5 | N | P |
|---|---|---|---|---|---|---|---|---|---|---|---|---|---|
| **benzoic acid derivatives** | | | | | | | | | | | | | |
| 1 | **gallic acid** | 2.39 | $C_7H_5O_5^-$ | 169.01425 | 169.01421 | 0.24 | 84(3), 123(8), 124(5), **125**(100), 126(8) | 69(49), 79(8), 81(93), 83(53), **97**(100) | ND | − | − | − | + |
| 2 | **protocatechuic acid**[a] | 4.50 | $C_7H_5O_4^-$ | 153.01933 | 153.01917 | 1.09 | 107(3), **109**(100), 110(8), 123(7) | 65(42), **81**(100), 91(68), 106(17) | ND | + | + | + | + |
| 3 | **P-hydroxybenzoic acid**[a] | 5.50 | $C_7H_5O_3^-$ | 137.02442 | 137.02425 | 1.25 | **93**(100), 94(6), 109(3) | ND | ND | + | + | + | + |
| **cinnamic acid derivatives** | | | | | | | | | | | | | |
| 4 | **3-O-caffeoylquinic acid** | 4.72 | $C_{16}H_{17}O_9^-$ | 353.08781 | 353.08614 | 4.73 | 135(6), 179(29), **191**(100), 192(4) | **85**(100), 93(62), 127(95), 173(73) | ND | + | + | − | − |
| 5 | **caffeoyl hexoside** | 5.26 | $C_{15}H_{17}O_9^-$ | 341.08781 | 341.08775 | 0.15 | 135(10), **179**(100), 180(9) | **135**(100) | **107**(100) | + | − | − | − |
| 6 | **5-O-caffeoylquinic acid**[a] | 5.36 | $C_{16}H_{17}O_9^-$ | 353.08781 | 353.0878 | 0.01 | 179(3), **191**(100) | **85**(100), 93(66), 127(89), 173(70) | **57**(100) | + | + | + | + |
| 7 | **coumaroyl hexoside** | 5.39 | $C_{15}H_{17}O_8^-$ | 325.09289 | 325.09286 | 0.1 | 119(9), **163**(100), 164(5), 289(18) | **119**(100) | ND | − | + | − | − |
| 8 | **caffeic acid**[a] | 5.89 | $C_9H_7O_4^-$ | 179.03498 | 179.03497 | 0.05 | 134(7), **135**(100) | 91(56), **107**(100), 117(16) | ND | + | + | + | − |
| 9 | **5-0-p-coumaroylquinic acid** | 6.40 | $C_{16}H_{17}O_8^-$ | 337.09289 | 337.09272 | 0.5 | 163(3), **191**(100), 192(4) | **85**(100), 93(55), 127(95), 173(68) | ND | − | + | − | − |
| 10 | **methyl 5-O-caffeoylquinate** | 6.51 | $C_{17}H_{19}O_9^-$ | 367.10346 | 367.10203 | 3.87 | 135(44), 161(11), **179**(100), 191(20) | **135**(100) | 79(53), **107**(100), 151(18) | − | − | − | + |
| 11 | **p-coumaric acid**[a] | 6.78 | $C_9H_7O_3^-$ | 163.04007 | 163.0397 | 2.27 | **119**(100), 120(4) | 91(5), **93**(100) | ND | + | + | + | − |
| 12 | **cinnamic acid** | 7.01 | $C_9H_7O_2^-$ | 147.04515 | 147.04454 | 4.16 | **103**(100) | ND | ND | + | + | + | − |

(Continued.)

**Table 3.** (Continued.)

| no. | compound name | $t_R$ (min) | molecular formula, [M−H]− | calculated mass, [M−H]− | exact mass, [M−H]− | Δ ppm | MS² fragments (% base peak) | MS³ fragments (% base peak) | MS⁴ fragments (% base peak) | H1 | H5 | N | P |
|---|---|---|---|---|---|---|---|---|---|---|---|---|---|
| 13 | ferulic acid[a] | 8.22 | $C_{10}H_9O_4^-$ | 193.05063 | 193.05036 | 1.41 | 134(34), **147**(100), 161(47), 178(15) | 101(13), 103(22), 111(8), **129**(100) | 55(9), 57(60), 73(3), **85**(100) | − | + | − | − |
| 14 | **benzyl caffeate** | 11.55 | $C_{16}H_{13}O_4^-$ | 269.08193 | 269.08163 | 1.13 | **134**(100), 135(5), 178(44), 225(8) | **106**(100), 108(16), 121(6), 150(29) | ND | + | + | + | − |
| 15 | **prenyl caffeate** | 11.56 | $C_{14}H_{15}O_4^-$ | 247.09758 | 247.09741 | 0.69 | 135(17), 161(3), **179**(100), 180(5) | **135**(100) | 65(8), 79(7), **107**(100), 117(5) | + | + | + | − |
| 16 | **cinnamyl caffeate** | 12.55 | $C_{18}H_{15}O_4^-$ | 295.09758 | 295.09756 | 0.09 | **134**(100), 178(89), 211(46), 251(49) | **106**(100), 109(12), 121(38) | ND | + | + | + | − |
| **flavan-3-ol monomers and dimers** | | | | | | | | | | | | | |
| 17 | **B type procyanidin dimer gallate** | 5.17 | $C_{37}H_{29}O_{17}^-$ | 745.14102 | 745.13843 | 3.48 | 423(12), 441(38), 467(21), **593**(100) | 289(8), 423(17), **441**(100), 467(30) | 153(36), 287(17), 289(21), **315**(100) | − | − | − | + |
| 18 | **methyl-B type prodelphinidin dimer** | 5.61 | $C_{31}H_{27}O_{13}^-$ | 607.14571 | 607.14453 | 1.94 | 287(45), 405(47), 423(15), **455**(100) | 315(8), **405**(100), 423(27), 437(82) | **243**(100), 283(4) | − | − | − | + |
| 19 | **epicatechin** | 5.93 | $C_{15}H_{13}O_6^-$ | 289.07176 | 289.07074 | 3.53 | 179(9), 205(28), **245**(100), 246(6) | 161(19), 187(25), 188(13), **203**(100) | 161(33), **175**(100), 185(21), 188(65) | − | − | − | + |
| 20 | **B type procyanidin dimer gallate isomer 1** | 6.18 | $C_{37}H_{29}O_{16}^-$ | 729.14611 | 729.14378 | 3.19 | 289(22), **407**(100), 559(73), 577(63) | 243(19), 255(21), 283(30), **285**(100) | 213(4), 241(4), **257**(100) | − | − | − | + |
| 21 | **(Epi)catechin gallate** | 6.84 | $C_{22}H_{17}O_{10}^-$ | 441.08272 | 441.08198 | 1.67 | 169(15), **289**(100), 303(3), 331(11) | 179(12), 205(34), 231(6), **245**(100) | 161(19), 187(20), 188(13), **203**(100) | − | − | − | + |
| 22 | **methyl-(epi)gallocatechin gallate** | 7.03 | $C_{23}H_{19}O_{11}^-$ | 471.09329 | 471.09266 | 1.34 | 169(15), **287**(100), 319(34), 439(42) | **125**(100), 161(9), 243(14), 245(3) | **57**(100) | − | − | − | + |
| 23 | **B type procyanidin dimer gallate isomer 2** | 7.19 | $C_{37}H_{29}O_{16}^-$ | 729.14611 | 729.14435 | 2.42 | **407**(100), 441(22), 559(59), 577(46) | 283(33), **285**(100), 297(35), 389(19) | 213(5), 241(3), **257**(100) | − | − | − | + |

(Continued.)

**Table 3.** (*Continued.*)

| no. | compound name | $t_R$ (min) | molecular formula $[M–H]–$ | calculated mass, $[M–H]–$ | exact mass, $[M–H]–$ | Δ ppm | MS² fragments (% base peak) | MS³ fragments (% base peak) | MS⁴ fragments (% base peak) | H1 | H5 | N | P |
|---|---|---|---|---|---|---|---|---|---|---|---|---|---|
| **flavones** | | | | | | | | | | | | | |
| **24** | **luteolin 6-C-hexoside** | 6.15 | $C_{21}H_{19}O_{11}^-$ | 447.09329 | 447.09291 | 0.84 | **327**(100), 328(11), 357(33), 358(6) | 284(6), **299**(100), 300(7) | 175(48), 213(66), **255**(100), 271(47) | – | + | – | + |
| **25** | **luteolin 8-C-hexoside** | 6.31 | $C_{21}H_{19}O_{11}^-$ | 447.09329 | 447.09177 | 3.38 | **327**(100), 357(35), 358(3), 369(5), 393(3) | 191(3), 255(3), 284(17), **299**(100) | 175(41), 213(61), 240(43), **255**(100) | – | + | – | – |
| **26** | **apigenin 8-C-hexoside (vitexin)**[a] | 6.65 | $C_{21}H_{19}O_{10}^-$ | 431.09837 | 431.09723 | 2.64 | **311**(100), 312(11), 341(15), 342(3) | **283**(100), 284(3) | 183(47), 211(29), 224(50), **239**(100) | – | + | – | + |
| **27** | **luteolin**[a] | 8.79 | $C_{15}H_9O_6^-$ | 285.04046 | 285.0397 | 2.68 | 151(39), 199(86), **241**(100), 243(60) | 185(15), **197**(100), 199(80), 213(58) | 152(16), 155(13), **169**(100), 179(15) | + | + | + | – |
| **28** | **apigenin**[a] | 9.65 | $C_{15}H_9O_5^-$ | 269.04555 | 269.0448 | 2.79 | 149(8), 151(5), 201(6), **225**(100), 226(15) | 181(13), 183(5), 197(3), **207**(100) | ND | + | + | + | – |
| **29** | **chrysoeriol** | 10.14 | $C_{16}H_{11}O_6^-$ | 299.05611 | 299.05609 | 0.07 | **284**(100), 285(13) | 227(8), **255**(100), 256(10) | 187(8), 211(92), 213(10), **227**(100) | + | + | + | – |
| **30** | **genkwanin**[a] | 10.59 | $C_{16}H_{11}O_5^-$ | 283.0612 | 283.06111 | 0.32 | 211(17), 239(82), 240(31), **268**(100) | 211(17), **239**(100), 240(30) | 195(73), **211**(100), 239(4) | + | – | – | – |
| **31** | **tricin** | 11.22 | $C_{17}H_{13}O_7^-$ | 329.06668 | 329.06659 | 0.28 | **314**(100), 315(11) | 271(7), 285(3), **299**(100) | 243(5), 255(4), **271**(100) | + | + | + | – |
| **32** | **chrysin**[a] | 11.75 | $C_{15}H_9O_4^-$ | 253.05063 | 253.05056 | 0.3 | 207(34), **209**(100), 210(13), 211(16) | 165(53), 167(16), 180(88), **181**(100) | **139**(100), 152(10), 153(99), 156(6) | + | + | + | – |
| **33** | **acacetin**[a] | 12.39 | $C_{16}H_{11}O_5^-$ | 283.0612 | 283.06116 | 0.11 | **268**(100), 269(10) | 211(11), **239**(100), 240(18) | 195(62), **211**(100), 239(3) | + | + | + | – |
| **flavonols** | | | | | | | | | | | | | |
| **34** | **myricetin** | 6.23 | $C_{15}H_9O_8^-$ | 317.03029 | 317.03015 | 0.43 | 163(14), **191**(100), 207(23), 299(31) | 135(5), **163**(100) | 91(4), 107(24), 119(56), **135**(100) | + | – | – | + |

(*Continued.*)

**Table 3.** (*Continued.*)

| no. | compound name | $t_R$ (min) | molecular formula [M−H]− | calculated mass [M−H]− | exact mass, [M−H]− | Δ ppm | MS² fragments (% base peak) | MS³ fragments (% base peak) | MS⁴ fragments (% base peak) | H1 | H5 | N | P |
|---|---|---|---|---|---|---|---|---|---|---|---|---|---|
| 35 | quercetin 3-O-(6''-rhamnosyl)-hexoside (Rutin)[a] | 6.51 | $C_{27}H_{29}O_{16}^-$ | 609.14611 | 609.14385 | 3.71 | 179(3), 300(31), **301**(100), 343(6) | 151(75), **179**(100), 271(47), 273(19) | **151**(100) | + | + | + | + |
| 36 | quercetin 3-O-galactoside[a] | 6.77 | $C_{21}H_{19}O_{12}^-$ | 463.0882 | 463.08774 | 1 | 300(36), **301**(100), 302(13) | 151(79), **179**(100), 255(25), 271(34) | **151**(100) | + | + | − | + |
| 37 | quercetin 3-O-rhamnoside[a] | 7.27 | $C_{21}H_{19}O_{11}^-$ | 447.09329 | 447.09261 | 1.51 | 300(21), **301**(100), 302(8) | 151(83), **179**(100), 255(27), 271(34) | **151**(100) | + | + | + | + |
| 38 | kaempferol 3-O-rhamnoside | 7.78 | $C_{21}H_{19}O_{10}^-$ | 431.09837 | 431.09787 | 1.17 | 255(6), 284(49), **285**(100), 286(15) | 229(30), **255**(100), 256(62), 257(80) | 210(5), 211(59), 212(4), **227**(100) | + | − | + | + |
| 39 | quercetin[a] | 8.85 | $C_{15}H_9O_7^-$ | 301.03538 | 301.03516 | 0.72 | 151(82), **179**(100), 257(12), 273(15) | **151**(100) | 63(4), 65(3), 83(13), **107**(100) | + | + | + | + |
| 40 | quercetin 3-methyl ether | 9.16 | $C_{16}H_{11}O_7^-$ | 315.05103 | 315.05066 | 1.16 | **300**(100), 301(10) | 254(9), 255(52), **271**(100), 272(6) | 199(16), 215(18), 227(67), **243**(100) | + | + | + | − |
| 41 | kaempferol[a] | 9.82 | $C_{15}H_9O_6^-$ | 285.04046 | 285.04022 | 0.85 | 151(71), 185(83), **229**(100), 239(78) | **185**(100), 187(41), 201(93), 211(43) | 142(91), **157**(100), 167(16) | + | + | + | + |
| 42 | herbacetin 8-methyl ether | 9.82 | $C_{16}H_{11}O_7^-$ | 315.05103 | 315.05089 | 0.43 | **300**(100), 301(10) | 255(59), 256(56), 271(23), **272**(100) | 137(24), 166(51), 216(30), **244**(100) | + | + | + | − |
| 43 | isorhamnetin | 10.01 | $C_{16}H_{11}O_7^-$ | 315.05103 | 315.05092 | 0.33 | **300**(100), 301(11) | **151**(100), 227(49), 271(96), 272(76) | 83(6), **107**(100) | + | + | + | − |
| 44 | dimethyl quercetin | 10.32 | $C_{17}H_{13}O_7^-$ | 329.06668 | 329.0665 | 0.52 | **314**(100), 315(11) | 271(3), **299**(100) | 227(6), 243(5), 255(9), **271**(100) | + | + | + | − |
| 45 | rhamnetin | 10.85 | $C_{16}H_{11}O_7^-$ | 315.05103 | 315.05081 | 0.69 | **165**(100), 193(33), 287(15), 300(44) | 65(24), 91(15), 97(53), **121**(100), 150(44) | 89(28), **91**(100), 93(20), 106(17) | + | + | − | − |

(*Continued.*)

**Table 3.** (Continued.)

| no. | compound name | $t_R$ (min) | molecular formula, [M−H]− | calculated mass, [M−H]− | exact mass, [M−H]− | Δ ppm | MS² fragments (% base peak) | MS³ fragments (% base peak) | MS⁴ fragments (% base peak) | H1 | H5 | N | P |
|---|---|---|---|---|---|---|---|---|---|---|---|---|---|
| 46 | **galangin**[a] | 11.97 | $C_{15}H_9O_5^-$ | 269.04555 | 269.04486 | 2.55 | 197(91), 198(50), **213**(100), 227(89) | 141(20), 169(95), **185**(100), 195(36) | **141**(100), 143(54), 157(31), 158(4) | + | + | + | − |
| 47 | **kaempferide**[a] | 12.03 | $C_{16}H_{11}O_6^-$ | 299.05611 | 299.05597 | 0.47 | 165(6), 271(5), **284**(100), 285(20) | **151**(100), 164(24), 228(22), 255(20) | 63(3), 65(3), 83(12), **107**(100), 122(5) | + | + | + | − |
| **flavanonols** | | | | | | | | | | | | | |
| 48 | aromodedrin | 8.01 | $C_{15}H_{11}O_6^-$ | 287.05611 | 287.05603 | 0.27 | 243(13), **259**(100), 260(12), 269(5) | 125(50), 151(18), 173(31), **215**(100) | 158(12), 172(30), **173**(100), 200(16) | + | + | + | − |
| 49 | pinobanksin 5-methyl ether | 9.16 | $C_{16}H_{13}O_5^-$ | 285.07685 | 285.07666 | 0.66 | 239(23), 252(16), **267**(100), 268(10) | 223(5), 224(8), 239(3), **252**(100) | 180(12), 208(70), 223(25), **224**(100) | + | + | + | − |
| 50 | pinobanksin | 9.94 | $C_{15}H_{11}O_5^-$ | 271.0612 | 271.06064 | 2.05 | 151(10), 197(18), 225(28), **253**(100) | 165(14), 181(27), 209(61), **225**(100) | 157(20), 181(27), 183(6), **197**(100) | + | + | + | − |
| 51 | pinobanksin 3-acetate | 12.02 | $C_{17}H_{13}O_6^-$ | 313.07176 | 313.07169 | 0.24 | **253**(100), 254(11), 271(13) | 165(12), 181(16), **209**(100), 211(13) | 153(24), 165(80), 180(96), **181**(100) | + | + | + | − |
| 52 | pinobanksin 3-propanoate | 12.92 | $C_{18}H_{15}O_6^-$ | 327.08741 | 327.08705 | 1.12 | **253**(100), 254(11), 271(5) | 165(14), 181(17), **209**(100), 211(11) | 165(48), 167(14), 180(47), **181**(100) | + | − | − | − |
| 53 | pinobanksin 3-butyrate | 13.78 | $C_{19}H_{17}O_6^-$ | 341.10306 | 341.10263 | 1.28 | **253**(100), 254(11) | 165(10), 181(18), **209**(100), 211(15) | 165(32), 167(13), 180(26), **181**(100) | + | + | − | − |
| 54 | pinobanksin 3-pentanoate | 14.59 | $C_{20}H_{19}O_6^-$ | 355.11871 | 355.11838 | 0.94 | **253**(100), 254(10), 291(11), 309(4) | 165(11), 181(18), 185(10), **209**(100) | 153(25), 165(58), 180(61), **181**(100) | + | + | + | − |
| **flavanone** | | | | | | | | | | | | | |
| 55 | **pinocembrin**[a] | 11.87 | $C_{15}H_{11}O_4^-$ | 255.06628 | 255.06512 | 4.55 | 151(32), 187(14), 211(34), **213**(100) | 145(18), 169(23), 184(3), **185**(100) | 115(19), 141(87), **143**(100), 157(18) | + | + | + | − |

[a]confirmed using available standards.

Ölschläger *et al*. [13] reported the presence of galloylated flavan 3-ols in buckwheat seeds, underscoring their absence in other crops. The presence of epicatechin ($m/z$ 289) and (epi)catechin gallate ($m/z$ 441) is characteristic for buckwheat [12–17,29]. Their presence in pollen has been confirmed among many other components, in particular methyl-B type prodelphinidin dimer ($m/z$ 607), methyl-(epi)gallocatechin gallate ($m/z$ 471) and B type procyanidin dimer gallate ($m/z$ 745) with its two isomers ($m/z$ 729).

UHPLC LTQ OrbiTrap analysis based on the accurate mass measurement and MS⁴ fragmentation provided efficient identification of compounds. In order to increase the advancement of these results, the quantification of target phenolic compounds was also performed. With a UHPLC-QqQ-MS/MS technique, a total of 31 phenolic compounds was quantified in buckwheat honey samples from Serbia and Poland, and the results are presented in the electronic supplementary material, table S3. Obtained results for tested Serbian buckwheat honey were very similar. Dominant content was found for propolis-derived flavonoids [51] (chrysin and pinocembrin), with average values higher than other authors reported [4,52,53]. The dominant contribution was also attributed to *p*-coumaric acid and *p*-hydroxybenzoic acid as others discovered [4,5,7,9] and suggested as markers for the botanical origin of buckwheat honey [4,5].

The higher average value of a total identified phenolics in honey from Poland, in comparison to Serbian honey, was attributed to quercetin and rutin, mostly owing to their markedly high amounts only in the sample H5 (26.40 and 7.99 mg kg$^{-1}$, respectively). The obtained values present 27.54% and 8.33%, respectively, of the total phenolic compounds in this sample. Many authors [4,7,52,53] reported lower amount of quercetin (known as pollen-nectar flavonoid [51]) in buckwheat honey. Unexpectedly, rutin was not found in each analysed buckwheat honey sample. Additionally, a notably higher amount was quantified in sample H5. Although rutin has been found in high quantities in different parts of buckwheat [10,11], it was not always present in buckwheat honey [4].

In view of melissopalynological analysis, results of the under-representative *Fagopyrum* pollen portion should indicate more obvious differences in the qualitative (table 3) and the quantitative analysis (electronic supplementary material, table S2) of phenolic compounds. However, phenolic profiles of analysed buckwheat honey samples appeared very similar.

## 3.4. Results of antioxidant tests

Results of spectrophotometrically determined values TPC and RSA showed high antioxidant activity of these samples (electronic supplementary material, table S2). Obtained values for RSA were in the range 5.85–10.25%, and for TPC were from 437.7 to 721.0 mg GAE kg$^{-1}$ for Serbian and 711.9–1496.8 mg GAE kg$^{-1}$ for Polish analysed buckwheat honey samples. It can be seen that the Polish honey sample H5 showed a twofold higher value. This high value supported the former statement for buckwheat honey from Poland [7,8], China [5,53] or Japan [9]. TPC and RSA values were correlated positively with each other, giving a correlation coefficient of 0.85 ($p < 0.05$). This was in accordance with observations of other authors [5,8,52], which indicate that phenolic compounds predominately contributed to the antioxidant activity of honey [5].

## 4. Conclusion

A major contribution of the comprehensive analysis refers to the phenolic profiles of buckwheat honey samples, which have a different portion of *Fagopyrum* pollen. Considering pollen analysis of buckwheat honey samples, it was shown that the presence of *Fagopyrum* pollen grains in buckwheat honey samples from Poland was assigned as 'secondary pollen' (16–45%), and in Serbia buckwheat honey samples as 'important minor' pollen (3–15%). Although pollen analysis indirectly assesses botanical origin, it is recognized as a method of choice that is generally acceptable. However, for honey with low pollen content such as buckwheat honey, it is of interest to have an additional parameter for confirmation of botanical origin. This study obtained direct correlation among Serbian buckwheat honey samples that possess a similar frequency of buckwheat pollen particles, as well as with buckwheat pollen and nectar samples, which are of the same geographical origin and were harvested in the same season (in July 2017). The indirect correlation that this study provides refers to the comparison of Serbian and Polish buckwheat honey samples that were harvested under different conditions, the latter of which had notably higher levels of buckwheat pollen particles. UHPLC LTQ OrbiTrap MS analysis confirmed not only the similarity of buckwheat honey samples but also a good correlation with the buckwheat nectar sample. Analysis of phenolic compounds presented in the nectar and pollen sample

could provide valuable additional data on the determination of the botanical origin of honey. Finally, in all analysed samples, a high content was identified of phytochemicals of interest for the health promotion qualities of buckwheat honey.

Data accessibility. All data (electronic supplementary material, tables S1 and S2, figure S1) are accessible in the electronic supplementary material files of this manuscript.

Authors' contributions. M.N. carried out the experimental analysis, participated in the design of the study and drafted the manuscript; U.G. participated in the experimental analysis, helped in the design of the study and helped draft the manuscript; T.T. participated in the experimental analysis; N.H. participated in the experimental analysis; N.N. helped draft the manuscript and collected field data and samples; B.Š. participated in the experimental analysis and in drafting the manuscript; S.B. helped designed the study; L.j.I. helped designed the study; Ž.T. conceived of the study, designed the study, coordinated the study and helped draft the manuscript. All authors have read and agreed to the published version of the manuscript.

Competing interests. We have no competing interests.

Funding. This work has been supported by Ministry of Education, Science and Technological Development of Republic of Serbia (contract nos 451-03-68/2020-14/200051, 451-03-68/2020-14/200007, 451-03-68/2020-14/200168, 451-03-68/2020-14/200288 and 451-03-68/2020-14/200358, 451-03-68/2020-14/200116).

Acknowledgements. We thank Nevena Mihailović for proofreading the final version of the manuscript.

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
