## [Reviewer comments · Royal Society Open Science]

Review History

RSOS-201576.R0 (Original submission)

Review form: Reviewer 1

Is the manuscript scientifically sound in its present form?

Yes

Are the interpretations and conclusions justified by the results?

No

Is the language acceptable?

No

Do you have any ethical concerns with this paper?

No

Have you any concerns about statistical analyses in this paper?

No

Recommendation?

Major revision is needed (please make suggestions in comments)

Comments to the Author(s)

The manuscript reports on the polyphenolic profile of buckwheat honey, nectar and pollen. The authors used state-of-the-art analytical methods to study the similarity of the polyphenolic profile of honey with the ones of nectar and pollen. These results provide important information for the scientific field. However, there are some issues that need to be addressed in order to improve the manuscript.

1. The manuscript requires an improvement in terms of language (grammatical, linguistic and spelling).
2. Regarding the number of samples analysed, not taking into account the diversity due to i.e. year of harvest, conclusions need to be formulated with more caution. With this in mind, I suggest revision and reformulation of:
 - the last two sentences of the Summary;
 - the Conclusions.
3. The term "different levels of monoflorality" is unusual and vague. Basing on the criteria set, a sample of honey is declared as monofloral or polyfloral, respectively. It holds true that for certain types of honey the criteria needs to be extended to declare monoflorality. I suggest to revise and rephrase this term throughout the manuscript.
4. In Chapter 3.1 (Samples), please provide a short note on how the sample of nectar and pollen was collected and stored.
5. In the Conclusions, please reformulate the part: "However, in the case of discrepancy, ..." bearing in mind that melissopalynological, physico-chemical and sensory analysis is used to determin / confirm botanical origin (Directive on honey, 2001) and the fact that buckwheat honey has characteristic sensory properties that allow it to be distinguished from other types of honey.
6. I suggest to omit word "significant" when describing differences, since statistical analysis was not performed as well as it is not in a place with this number of samples.
7. In the supplementary material, in table S2, please check the sample labels for samples from Poland - there are two H5 labels and H6 is missing.

Review form: Reviewer 2

Is the manuscript scientifically sound in its present form?

No

Are the interpretations and conclusions justified by the results?

No

Is the language acceptable?

No

Do you have any ethical concerns with this paper?

No

Have you any concerns about statistical analyses in this paper?

No

Recommendation?

Major revision is needed (please make suggestions in comments)

Comments to the Author(s)

Dear authors,

This is an interesting manuscript. However I pointed some suggestions/modification in your manuscript.

I believe that can improve your work. I am attached the PDF file (Appendix A) with my observations.

Decision letter (RSOS-201576.R0)

Dear Professor Tešić:

Title: Polyphenolic profile of buckwheat honey, nectar and pollen
Manuscript ID: RSOS-201576

The editor assigned to your manuscript has now received comments from reviewers. We would like you to revise your paper in accordance with the referee and Subject Editor suggestions which can be found below (not including confidential reports to the Editor). Please note this decision does not guarantee eventual acceptance.

Please submit your revised paper before 30-Oct-2020. Please note that the revision deadline will expire at 00.00am on this date. If we do not hear from you within this time then it will be assumed that the paper has been withdrawn. In exceptional circumstances, extensions may be possible if agreed with the Editorial Office in advance. We do not allow multiple rounds of revision so we urge you to make every effort to fully address all of the comments at this stage. If deemed necessary by the Editors, your manuscript will be sent back to one or more of the original reviewers for assessment. If the original reviewers are not available we may invite new reviewers.

Royal Society of Chemistry
Thomas Graham House
Science Park, Milton Road
Cambridge, CB4 0WF

Royal Society Open Science - Chemistry Editorial Office

RSC Associate Editor:
Comments to the Author:
(There are no comments.)

RSC Subject Editor:
Comments to the Author:
(There are no comments.)

Reviewers' Comments to Author:
Reviewer: 1

Comments to the Author(s)

The manuscript reports on the polyphenolic profile of buckwheat honey, nectar and pollen. The authors used state-of-the-art analytical methods to study the similarity of the polyphenolic profile of honey with the ones of nectar and pollen. These results provide important information for the scientific field. However, there are some issues that need to be addressed in order to improve the manuscript.

1. The manuscript requires an improvement in terms of language (grammatical, linguistic and spelling).
2. Regarding the number of samples analysed, not taking into account the diversity due to i.e. year of harvest, conclusions need to be formulated with more caution. With this in mind, I suggest revision and reformulation of:
 - the last two sentences of the Summary;
 - the Conclusions.
3. The term "different levels of monoflorality" is unusual and vague. Basing on the criteria set, a sample of honey is declared as monofloral or polyfloral, respectively. It holds true that for certain types of honey the criteria needs to be extended to declare monoflorality. I suggest to revise and rephrase this term throughout the manuscript.
4. In Chapter 3.1 (Samples), please provide a short note on how the sample of nectar and pollen was collected and stored.
5. In the Conclusions, please reformulate the part: "However, in the case of discrepancy, ..." bearing in mind that melissopalynological, physico-chemical and sensory analysis is used to determin / confirm botanical origin (Directive on honey, 2001) and the fact that buckwheat honey has characteristic sensory properties that allow it to be distinguished from other types of honey.
6. I suggest to omit word "significant" when describing differences, since statistical analysis was not performed as well as it is not in a place with this number of samples.
7. In the supplementary material, in table S2, please check the sample labels for samples from Poland - there are two H5 labels and H6 is missing.

Reviewer: 2

Comments to the Author(s)
Dear authors,

This is an interesting manuscript. However I pointed some suggestions/ modification in your manuscript.

I believe that can improve your work. I am attached the PDF file with my observations.

Author's Response to Decision Letter for (RSOS-201576.R0)

See Appendix B.

Decision letter (RSOS-201576.R1)

Dear Professor Tešić:

Title: Polyphenolic profile of buckwheat honey, nectar and pollen
Manuscript ID: RSOS-201576.R1

It is a pleasure to accept your manuscript in its current form for publication in Royal Society Open Science. The chemistry content of Royal Society Open Science is published in collaboration with the Royal Society of Chemistry.

RSC Associate Editor
Comments to the Author:
(There are no comments.)

Reviewer(s)' Comments to Author:

Appendix A**ROYAL SOCIETY
OPEN SCIENCE****Polyphenolic profile of buckwheat honey, nectar and pollen**

Journal:	Royal Society Open Science
Manuscript ID	RSOS-201576
Article Type:	Research
Date Submitted by the Author:	04-Sep-2020
Complete List of Authors:	Nešović, Milica; Institute of General and Physical Chemistry Gašić, Uroš; University of Belgrade, Department of Plant Physiology Tosti, Tomislav; University of Belgrade Faculty of Chemistry Horvacki, Nikola; University of Belgrade Faculty of Chemistry, Innovation Center Škoparija, Branko; University of Novi Sad, BioSense Institute - Research Institute for Information Technologies in Biosystems Nedić, Nebojša; University of Belgrade, Faculty of Agriculture, Institute for zootechnics Blagojević, Stevan; Institute of General and Physical Chemistry Ignjatović, Ljubiša; University of Belgrade Faculty of Physical Chemistry Tešić, Živoslav; University of Belgrade Faculty of Chemistry,
Subject:	Analytical chemistry < CHEMISTRY
Keywords:	Buckwheat honey, botanical origin, polyphenol profile, flavan-3-ols, UHPLC MS OrbiTrap
Subject Category:	Chemistry

1
2
3 **Author-supplied statements**
4

5 Relevant information will appear here if provided.
6

7
8 ***Ethics***
9

10 *Does your article include research that required ethical approval or permits?:*

11 This article does not present research with ethical considerations
12

13 *Statement (if applicable):*

14 CUST_IF_YES_ETHICS :No data available.
15
16

17 ***Data***
18

19 *It is a condition of publication that data, code and materials supporting your paper are made publicly*
20 *available. Does your paper present new data?:*

21 Yes
22

23 *Statement (if applicable):*

24 Our work contains new data, but does not use data/models published elsewhere.
25
26

27 ***Conflict of interest***
28

29 I/We declare we have no competing interests
30

31 *Statement (if applicable):*

32 The authors declare no conflicts of interest.
33
34

35 ***Authors' contributions***
36

37 This paper has multiple authors and our individual contributions were as below
38

39 *Statement (if applicable):*

40 CUST_AUTHOR_CONTRIBUTIONS_TEXT :No data available.
41
42
43
44
45
46
47
48
49
50
51
52
53
54
55
56
57
58
59
60

Polyphenolic profile of buckwheat honey, nectar and pollen

Milica Nešović¹, Uroš Gašić², Tomislav Tosti³, Nikola Horvacki⁴, Branko Šikoparija⁵, Nebojša Nedić⁶, Stevan Blagojević¹, Ljubiša Ignjatović⁷ and Živoslav Tešić^{3*}

¹*Institute of General and Physical Chemistry, University of Belgrade, Studentski trg 12-16, 11158 Belgrade, Serbia;*

²*Department of Plant Physiology, Institute for Biological Research "Siniša Stanković", National Institute of Republic of Serbia, University of Belgrade, Bulevar despota Stefana 142, 11060 Belgrade, Serbia;*

³*University of Belgrade – Faculty of Chemistry, Studentski trg 12-16, P. O. Box 51, 11158 Belgrade, Serbia;*

⁴*Innovative Center, University of Belgrade - Faculty of Chemistry, Studentski trg 12-16, 11158 Belgrade; Serbia;*

⁵*University of Novi Sad, BioSense Institute - Research Institute for Information Technologies in Biosystems, Trg Dr Zorana Đinđića 1, 21000 Novi Sad, Serbia;*

⁶*Faculty of Agriculture, Institute for zootehnics, University of Belgrade, Nemanjina 6, 11080 Belgrade-Zemun, Serbia;*

⁷*University of Belgrade, Faculty of Physical Chemistry, Studentski trg 12-16, 11000 Belgrade, Serbia;*

Keywords: Buckwheat honey, botanical origin, polyphenol profile, flavan-3-ols, UHPLC MS OrbiTrap.

1. Summary

The focus of research in recent years is the comparability of honey as the final product of bees, and pollen and nectar of the plant from which honey originates, as the main food source for bees. Buckwheat honey is recognized as a nutritionally valuable product, which provide a scientifically proven health benefit and confirmed it as a functional food. The quality of this type of honey is attributed to high levels of phytochemicals in buckwheat. The purpose of this study was the examination of coherence between buckwheat honey and buckwheat nectar and pollen, as well as simultaneous investigation of its origin and the chemical analysis. The phenolic profile of buckwheat pollen showed a lower number of flavonoids and phenolic acids than in nectar and honey samples but confirmed the presence of well-known polyphenols derived from the buckwheat plant. Significant difference was found on the presence of (epi)catechin units, its galloylated derivatives and procyanidin dimers, which were not present in honey. An important achievement of this study refers on the obtained good correlation of polyphenols of buckwheat nectar and honey that contains a low amount of buckwheat pollen particles. Additionally, this provides new possibilities for predicting the botanical origin of honey.

2. Introduction

Buckwheat is a pseudo-cereal from Polygonaceae family, and as a multipurpose plant it is recognizes as an essential functional food [1]. Common buckwheat (*Fagopyrum esculentum* Moench) is a reliable and high-

* Živoslav Lj. Tešić (ztesic@chem.bg.ac.rs).

†Present address: Studentski trg 12-16, Belgrade 11158, Serbia.

yielding honey plant [2]. A large number of flowers that bloom for a long time contribute to the buckwheat fields being a valuable and suitable bee pasture.

Buckwheat honey is highly valued, with specific organoleptic properties, such as strong flavor, characteristic color, pungent taste and aroma of molasses, but not preferred by consumers [3]. Regardless of estimation by consumers, buckwheat honey has a very high nutritional value with beneficial antioxidant [4,5] and anti-inflammatory effects [6]. As compared with other types of honey, buckwheat honey showed a higher content of minerals, sugars, phenolic compounds [7-9].

Buckwheat is a good source of nutrients and bioactive compounds [1,2] which are significantly responsible for the antioxidant activity it possesses. A high content of rutin was noted in all parts of buckwheat [10-13]. Monomeric flavan-3-ols (catechin and epicatechin) and procyanidins were found in buckwheat extracts [12-17] and proposed as buckwheat markers. As concerns the buckwheat honey *fingerprnt*, many authors suggested the dominant presence of *p*-coumaric acid and *p*-hydroxybenzoic acid [4,5]. Monofloral honey should predominantly contain pollen from plant species that has declared that honey type. According to Beckh et al. [18], buckwheat honey must contain over 30 % of buckwheat pollen. Such buckwheat monofloral honey is relatively rare. It is mostly found in countries with high rates of buckwheat production, such as China, Russia, Ukraine, then Poland, while in Serbia buckwheat fields are on a significantly smaller area [19]. Moreover, published data show high variability of level of *Fagopyrum* pollen [3-5,8,9,20,21], whereas only Polish Standard [20] still define buckwheat honey with over 45 % of pollen. As there is the appearance of different levels of "monoflorality" for buckwheat honey [4], it is significant to find additional phytochemicals that could be useful for establishing the accurate classification of this type of honey.

The objective of this study was to characterize buckwheat honey samples from Serbia and Poland based on analysis of phenolic compounds, as well as antioxidant tests (total phenolic content (TPC) and radical scavenging activity (RSA)), pollen analysis and physicochemical parameters (moisture content, electrical conductivity and sugar profile). In addition to the generally accepted pollen analysis for determining botanical origin of honey, analysis of polyphenols could be a method of choice, especially for honey with a low content of pollen. In order to provide specific phenolic compounds in buckwheat honey, high-resolution mass spectrometry (HRMS) in combination with MSⁿ fragmentations was used. To obtain a more accurate correlation of phenolic compounds with botanical origin of honey, the identification of phenolic compounds in buckwheat nectar and buckwheat pollen sample was also performed. By correlating the polyphenolic profile of buckwheat honey, nectar and pollen, we wanted to obtain additional important data for the assessment of the botanical origin of honey with low pollen content. For this purpose, four samples of honey from Serbia with a low content of buckwheat pollen particles declared by beekeepers as buckwheat honey, and two buckwheat honey samples from Poland with high content of buckwheat pollen were analyzed. With this in mind, two carefully selected samples of honey, containing the low and high percentage of buckwheat pollen, were more detailed studied for polyphenol profile.

3. Materials and Methods

3.1. Samples

For this study, six buckwheat honey samples were analysed (Table 1). Four honey samples (assigned as H1-H4) were provided by Serbian beekeepers, during 2017, which hives were settled in a small geographical area (of about 2500 km²) in Western Serbia. Buckwheat honey samples H5 and H6 were received from the commercial market in Poland in 2016. Buckwheat nectar and pollen were collected directly from flowers of buckwheat, cultivated in Serbia within the previously mentioned area.

3.2. Chemical and materials

Folin-Ciocalteu's (FC) reagent, gallic acid, sodium carbonate, 2,2-diphenyl-1-picrylhydrazyl (DPPH), Trolox standard, sodium hydroxide, sodium acetate, and all chemicals for chromatographic analysis of polyphenols (hydrochloric acid, acetonitrile, formic acid, and phenolic standards) were purchased from Sigma Aldrich (Steinheim, Germany). Sugar standards (glucose, fructose, sucrose, maltose, isomaltose, trehalose, turanose, melibiose and melezitose) were supplied from Tokyo Chemical Industry, Europe (Zwijndrecht, Belgium).

1
2
3
4
5
6
7
8
9
10
11
12
13
14
15
16
17
18
19
20
21
22
23
24
25
26
27
28
29
30
31
32
33
34
35
36
37
38
39
40
41
42
43
44
45
46
47
48
49
50
51
52
53
54
55
56
57
58
59
60

Ultrapure water (18 MΩcm) produced by TKA MicroPure water purification system (Thermo Fisher TKA, Niederelbert, Germany) was used for the preparation of all aqueous solutions. Syringe filters (25 mm, PTFE membrane 0.45 μm) supplied by Supelco (Bellefonte, PA, USA) was used for filtration of samples. Cartridges (Strata C18-E, 500 mg/3mL) for solid phase extraction (SPE) were obtained from Phenomenex (Torrance, CA, USA).

3.3. Melissopalynology analysis

Frequency of pollen was determined for all six honey samples. Pollen was extracted and analysed following the Harmonized Methods of Melissopalynology [22]. Microscopic slides were scanned until a minimum of 500 pollen grains was counted and identified using referent slides and pollen identification atlases [23]. Honey dew elements i.e. algae, fungal spores and hyphae, anemophilous pollen and pollen of nectarless plants [24] was also counted. Relative frequency of identified pollen was calculated [25] and the frequency classes [26] were reported including the values after nectarless species are excluded from the analysis. Honeydew contribution is evaluated as the ratio between the number of honeydew elements and pollen of nectariferous plants according to published scale [26].

3.4. Determination of physicochemical parameters

According to the method described by Bogdanov et al. [27], moisture content, electrical conductivity, and sugars content were determined. Content of sugars was analysed using high-performance anion exchange chromatography with a pulsed amperometric detector (HPAEC/PAD) (Dionex ICS 3000) with previously described procedure for preparation of sugar standards [28]. The refractive index of honey samples was measured using an Abbe-type refractometer (Atago RX refractometer, Japan) at the temperature of 20 °C. After using the Chataway's table correction on these values, the results of the moisture content of honey samples were obtained. Conductivity meter (WTW Golden lab & engineering, Cond 7110, Inolab) with conductivity cell with its constant of 0.108 cm⁻¹, was used for measuring the electrical conductivity of 20 % (w/v) aqueous solution of honey.

3.5. Phenolic profile

The preparation of samples and standard solutions for phenolic profile analysis was achieved by following the previously described procedure by Gašić et al. [28]. Stock solutions of available phenolic standards were prepared in methanol in a concentration range of 0.025 to 2.000 mg/L. Isolation of phenolic compounds from samples was carried out using SPE.

Qualitative analysis of phenolic compounds was carried out by an Accela 600 UHPLC system connected to LTQ OrbiTrap mass spectrometer (Thermo Fisher Scientific, Bremen, Germany). Chromatographic and spectrometric conditions, as well as instruction for fragmentation study, were previously described by Gašić et al. [28]. The presence of some phenolic compounds was confirmed using available standards, while the other was identified by examination of previously published MS fragmentation data. Tentative identification of some phenolic compounds in the absence of standards was achieved by HRMS and MS⁴ fragmentation.

Quantitative analysis of phenolic compounds was performed using Dionex ICS 3000 liquid chromatography system connected to TSQ Quantum Access Max triple-quadrupole mass spectrometer (Thermo Fisher Scientific, Basel, Switzerland) followed by chromatographic and mass spectrometry conditions such as those previously used by Gašić et al. [29]. Quantification of phenolic compounds was based on direct comparison with available standards and results were expressed as mg/kg.

3.6. Antioxidant tests

Spectrophotometric determinations of TPC and RSA were performed in the reaction with FC reagent and DPPH, respectively, by following the procedures previously described by Nešović et al. [30]. UV/Vis spectrophotometer (GBC UV/Visible Cintra 6, Australia) was used for recording of wavelengths.

4. Results and Discussion

4.1. Melissopalynology analysis

The dominant presence of pollen grains from nectarless plant species (e.g. *Fraxinus americana*) was noticed in analysed buckwheat honey samples (Table S1). This is ornamental species that flowers before buckwheat, during April and May and if its pollen is collected in large quantities it can “contaminate” nectar stored in the comb. As bees collect pollen to obtain many proteins, minerals, and lipids they cannot find in nectar [31], they often visit different types of plants to collect pollen with the appropriate composition for them. Therefore, the pollen grains present in honey do not necessarily originate from the same species as nectar, as also pointed out by others [32].

After removal of pollen from nectarless species, qualitative melissopalynological analysis identified *Fagopyrum* pollen at levels of 4.05-10.93 % in analysed Serbian buckwheat honey, compared to 40.83 % and 24.29 % in Polish honey assigned as H5 and H6, respectively. *Fagopyrum* pollen was important minor pollen in Serbian (3-15 %) and secondary pollen (16-45 %) in Polish honey samples declared as buckwheat (Table 1). Besides *Fagopyrum*, pollen from nectariferous plants belonging to Boraginaceae, Rosaceae, Fabaceae, Brassicaceae, Asteraceae, Hypericaceae, Violaceae and Malvaceae families, was identified in both Serbian and Polish honey samples. None of the identified pollen can be classified as predominant. Pollen grains from *Astragalus* and *Amorpha* types (Fabaceae), *Echium* (Boraginaceae) and *Filipendula* (Rosaceae) were found as secondary pollen in Serbian samples. Buckwheat honey from China was also contained a high content of pollen grains from the *Astragalus* [33]. In analysed Polish buckwheat honey samples, *Fagopyrum* was secondary pollen together with *Rubus*, *Brassica napus* and other Brassicaceae.

A wide range of pollen grains was identified in Polish buckwheat honey and presence of notable amounts of *Brassica napus* and *Phacelia* pollen, so as some *Helianthus* pollen indicates bees were foraged in more agriculture area that it is the case for Serbian samples. The other pollen types recorded in Polish honey samples correspond to the spectrum discovered by others [20]. However, the quantity of that pollen in our samples could not influence its classification.

As it appears from the results of melissopalynological analysis, our honey samples declared as buckwheat have a widespread range of *Fagopyrum* pollen. In view of the fact there are different types of “monoflorality” of buckwheat honey [4], Polish honey samples have appropriate declaration, while Serbian honey samples could be potentially considered as buckwheat honey. These results partially confirmed observations of other authors who reported high variability of *Fagopyrum* pollen in buckwheat honey, from 0.3 to 86 % [3-5,8,9,20,21,34].

Generally, buckwheat pollen is not expected to be under-represented in honey [35]. The size of *Fagopyrum esculentum* Moench pollen ranges 26-50 μm indicating it could be easier filtered out during nectar transport especially if sources are distant from the hive. However, high variability in contribution of *Fagopyrum* pollen in honey could be due the fact there are two types of flowers on buckwheat, which do not have same flow of pollen and nectar [36]. Different pollen flow for distylous species was already noted for the plant from Rubiaceae family [37]. Bees collect pollen to obtain many proteins, minerals, and lipids they cannot find in nectar [31]. Therefore, during *Fagopyrum* nectar flow, bees are forced to visit various sources to satisfy their feeding requirements. In addition, bees also pick up pollen by touching stamens of various flowers. Consequently, the pollen grains present in honey do not necessarily originate from the same species as nectar [32]. As a result, honey can occur with under-representative pollen, meaning that frequency of pollen in honey does not correlate with the corresponding nectar contribution [24]. Furthermore, in countries with low buckwheat production, such as Serbia, there are small areas with cultivated buckwheat making the source of food for bees limited.

4.2. Physicochemical analysis

Comparing results of physicochemical parameters for analyzed Serbian and Polish buckwheat honey sample (Table 2), there were no significant differences. The obtained values of moisture content are below than 17.1 % and in accordance with the regulations for honey [38] as well as the content of monosaccharides (higher than 60 g/100 g) and sucrose (less than 3 g/100 g). The values for electrical conductivity of Serbian honey were between 0.32-0.43 mS/cm which were very similar to published data [34,39], while for Polish honey were for trace lower, 0.21 mS/cm and 0.28 mS/cm. Another published values were higher, such as for buckwheat honey from Poland, i.e. in range 0.40-0.53 mS/cm [20], as well for buckwheat honey from China, i.e. 0.35-0.63 mS/cm [5].

The sugar profile of buckwheat honey besides dominant fructose and glucose, contain oligosaccharides typical for blossom honey such as turanose, maltose and trehalose [40]. Determined values of sums of sugars (from 67.00-71.53 g/100 g) were lower than other authors reported [4,41]. Melezitose was found in our six analyzed buckwheat honey samples, while others [4,41] did not found it.

Our results of physicochemical parameters confirmed observations of Popek et al. [39], who proposed distinguishing of buckwheat honey based on Decision Tree developed classification model by following parameters: electrical conductivity ≤ 0.45 mS/cm, sucrose ≤ 3.105 g/100 g and reducing sugars ≤ 78.525 g/100 g.

4.3. Phenolic profile

Comparing mass chromatograms with available standards or in the absence of standards, with published MS fragmentation pathways [12-15,17,28], it was identified 55 phenolic compounds. Results of the qualitative analysis of two buckwheat honey with low and high content of pollen particles (H1 and H5), as well as buckwheat nectar (N) and pollen (P), are presented in Table 3. Analyzed buckwheat samples contained a notable number of phenolic compounds, identified with an order which was showed to be as Polish buckwheat honey (41) > Serbian buckwheat honey (40) > buckwheat nectar (33) > buckwheat pollen (21). The most common compounds, according to the number identified, were flavonoids, followed by phenolic acids and flavonoid glycosides (Table 3).

Phenolic acids were identified as free acids and in form of hexosides and esters. Two derivatives of caffeoylquinic acid (compounds 4 and 6) were found in honey samples, while 5-O derivative (chlorogenic acid) was found only in nectar and pollen. Two more quinic acid esters were found in the tested samples; compound 9 (5-O-*p*-coumaroylquinic acid) only in buckwheat honey from Serbia and compound 10 (methyl 5-O-caffeoylquinic acid) only in pollen sample. Identification of all these quinic acid derivatives, including esterification position, was in accordance with previously published MS and chromatographic data [42].

Propolis-derived flavonoids (chrysin, pinocembrin, pinobanksin and galangin) were found in both honey and buckwheat nectar samples. As nectar is the main source of food for bees, it is natural for the phenolic profile to be similar to honey. The phenolic compounds present in honey came from buckwheat nectar but also from many other plant species that bees are visiting, which resulting presence of more other phenolic compounds in honey. Compound 42 was not identified only in pollen sample, so it can be said to originate from nectar or propolis. Glycoside derivatives of this compound have previously been found in *Brassica napus* honey [43], while the detailed fragmentation pathway of compound 42 was shown in our previous study [44]. Likewise, pinobanksin derivatives (compounds 49, 51-54) were not found only in the tested pollen sample, but they are known to be propolis-derived compounds and their fragmentation is well described in the literature [45]. There were noted phenolic compounds that differentiate buckwheat nectar from corresponding honey, but they did not include flavonoid glycosides (Table 3). However, flavonoid glycosides are considered to dominate in honey due to hydrolysis in the presence of bee-enzymes [4]. Nevertheless, three flavonol 3-O-glycosides (quercetin 3-O-rhamnoside, quercetin 3-O-(6"-rhamnosyl)-glucoside (rutin), kaempferol 3-O-rhamnoside) were common for nectar and Serbian honey, as well as pollen. Quercetin 3-O-galactoside (hyperoside) was not identified only in buckwheat nectar sample. Glycosylation position of all identified flavonol glycosides (compounds 35-38) were confirmed by the presence of the MS² base peak at mass of aglycone and high intensity of its radical ion (*m/z* 300 for quercetin and *m/z* 284 for kaempferol) [46]. Generally, rutin also appears in different parts of buckwheat [12-16]. The presence of kaempferol rhamnosides was identified in both nectar and honey samples from *Robinia pseudacacia* [47]. Later, it was reported the presence of kaempferol rhamnosides and rhamnosyl-glycosides, through pollen, nectar, and honey from *Diploptaxis tenuifolia* [48].

Aglycones are more potent antioxidants than corresponding glycosides, but the glucose portion can enhance the bioavailability of compounds [49]. Isomers of flavone 6-C-glycosides and flavone 8-C-glycosides were differentiated in buckwheat sprouts [50]. In our study, luteolin 6-C-hexoside and luteolin 8-C-hexoside were found in buckwheat honey from Poland.

Considering a variety of identified phenolic compounds in analyzed buckwheat samples, it was found a smaller number of phenolic acids in pollen, as well as flavonoids, of which the group of *O*-methylated flavonoids generally were not present in pollen. Of 21 phenolic compounds found in pollen, only myricetin (m/z 317) was common to pollen and honey from Serbia, which is pollen-nectar-derived flavonoid [51]. It was published the presence of myricetin in buckwheat honey [45], and in buckwheat flours (in a bound form) [15]. Furthermore, it was reported [34] its presence in many honey types as well in buckwheat honey, while in their earlier study the excess of myricetin in comparison to the rest of the flavonoids proposed as a chemical marker of the authenticity of heather honey [7].

Phenolic compounds that were present only in pollen sample, were mostly (epi)catechin units, its galloylated derivatives, and procyanidin dimers, which have already been considered as buckwheat markers [12-17,28]. It was reported potential health effects of procyanidins B type with its higher content in buckwheat than in cereals (barley and spelt) [17]. Additionally, procyanidins increase antioxidant effect against superoxide anions more than monomeric flavonoids [49]. Ölschläger et al. [13] were reported the presence of galloylated flavan 3-ols in buckwheat seeds, underscoring their absence in other crops. The presence of epicatechin (m/z 289) and (epi)catechin gallate (m/z 441) is characteristic for buckwheat [12-17,28]. Their presence in pollen has been confirmed among many other components, in particular methyl-B type prodelfinidin dimer (m/z 607), methyl-(epi)gallo catechin gallate (m/z 471), and B type procyanidin dimer gallate (m/z 745) with its two isomers (m/z 729).

UHPLC LTQ OrbiTrap analysis based on accurate mass measurement and MS⁴ fragmentation provided efficient identification of compounds. In order to increase advancement of these results, quantification of target phenolic compounds was also performed. With UHPLC-QqQ-MS/MS technique, total of 31 phenolic compounds was quantified in buckwheat honey samples from Serbia and Poland, and the results are presented in Table S3. Obtained results for tested Serbian buckwheat honey were very similar. Dominant content was found for propolis-derived flavonoids [51] (chrysin and pinocembrin), with average values higher than other authors reported [4,52,53]. The dominant contribution was also attributed to *p*-coumaric acid and *p*-hydroxybenzoic acid as other discovered [4,5,7,9] and suggested as markers for the botanical origin of buckwheat honey [4,5].

The higher average value of a sum of phenolics in honey from Poland, in comparison to Serbian honey, was attributed to quercetin and rutin, mostly due to its markedly high amounts only in sample H5 (26.40 and 7.99 mg/kg, respectively). Obtained values present 27.54 % and 8.33 %, respectively, of the sum of phenolic compounds in this sample. Many authors [4,7,52,53] reported lower amount of quercetin (known as pollen-nectar flavonoid [51]) in buckwheat honey. Rutin was not found in each analysed buckwheat honey samples, which was not expected. Additionally, a significant amount was quantified in sample H5. Although rutin has been found in high quantities in different parts of buckwheat [10,11], it was not always present in buckwheat honey [4].

In view of melissopalynological analysis, results of under-representative *Fagopyrum* pollen portion should indicate more obvious differences in the qualitative (Table 3) and the quantitative analysis (Table S2) of phenolic compounds. However, phenolic profiles of analysed buckwheat honey samples appeared as very similar.

4.4. Results of antioxidant tests

Results of spectrophotometrically determined values TPC and RSA showed high antioxidant activity of these samples (Table S2). Obtained values for RSA were in the range of 5.85-10.25 %, and for TPC were from 437.7-721.0 mg GAE/kg for Serbian and 711.9-1496.8 mg GAE/kg for Polish analysed buckwheat honey samples. It can be seen that the Polish honey sample H5 showed a two-fold higher value. This high value was supporting the former statement for buckwheat honey from Poland [7,8], China [5,53], or Japan [9]. TPC and RSA values were correlated positively with each other, giving correlation coefficient of 0.85 ($p < 0.05$). This was in accordance with observations of other authors [5,8,52], which indicate that phenolic compounds predominately contributed to the antioxidant activity of honey[5].

5. Conclusion

Major contribution of the comprehensive analysis refers to the phenolic profiles of buckwheat honey samples, which have a different portion of *Fagopyrum* pollen. Considering pollen analysis of buckwheat honey samples, it was shown that the presence of *Fagopyrum* pollen grains in buckwheat honey samples from Poland was assigned as “secondary pollen” (16-45 %), and in Serbia buckwheat honey samples as “important minor” pollen (3-15 %). Although pollen analysis indirectly assesses botanical origin, it is recognized as a method of choice that is generally acceptable. However, in the case of discrepancy, which is very noticeable for honey with low pollen content such as buckwheat honey, it is of interest to have an additional parameter for confirmation of botanical origin. UHPLC LTQ OrbiTrap MS analysis confirmed not only the similarity of buckwheat honey samples but also a good correlation with buckwheat nectar sample. Analysis of phenolic compounds presented in nectar and pollen sample provides valuable additional data on the determination of the botanical origin of honey. Furthermore, polyphenols of buckwheat pollen have shown a satisfactory correlation with honey, even in low-pollen honey samples. Additionally, all buckwheat honey samples showed high nutritional value, as well as the content of phytochemicals of interest for the health promotion of buckwheat honey.

Funding Statement

This work has been supported by Ministry of Education, Science and Technological Development of Republic of Serbia (Contract No: 451-03-68/2020-14/200051, 451-03-68/2020-14/200007, 451-03-68/2020-14/200168, 451-03-68/2020-14/200288 and 451-03-68/2020-14/200358).

Data Accessibility

Supplementary results (Tables S1 and S2) are uploaded as electronic supplementary material.

Competing Interests

We have no competing interests.

Authors' Contributions

MN, UG, TT, NH, and BŠ carried out the molecular lab work, participated in data analysis, carried out sequence alignments, participated in the design of the study and drafted the manuscript; NN collected field data; SB, LjI, and ŽT conceived of the study, designed the study, coordinated the study and helped draft the manuscript. All authors read and approved the manuscript before submission.

References

- Zhang ZL, Zhou ML, Tang Y, Li FL, Tang YX, Shao JR, Xue WT, Wu YM. 2012. Bioactive compounds in functional buckwheat food. *Food Res. Int.* **49**, 389-395. (doi: 10.1016/j.foodres.2012.07.035)
- Sikora V. 2013. Buckwheat as a honey plant, presented in part at *Proceedings of the XXXI advising beekeepers*, 45-60, ISBN 978-86-89519-00-6. (in serbian), Novi Sad, Serbia.
- Kortesniemi M, Rosenvald S, Laaksonena O, Vanaga A, Ollikkac T, Vene K, Yanga B. 2018. Sensory and chemical profiles of Finnish honeys of different botanical origins and consumer preferences. *Food Chem.* **246**, 351-359. (doi: 10.1016/j.foodchem.2017.10.069)
- Pasini F, Gardini S, Marcazzan GL, Caboni MF. 2013. Buckwheat honeys: screening of composition and properties. *Food Chem.* **141**(3), 2802-2811. (doi: 10.1016/j.foodchem.2013.05.102)
- Zhou J, Li P, Cheng N, Gao H, Wang B, Wei Y, Cao W. 2012. Protective effects of buckwheat honey on DNA damage induced by hydroxyl radicals. *Food Chem. Toxicol.* **50**, 2766-2773. (doi:10.1016/j.fct.2012.05.046)
- Borutinskaite V, Treigyte G, Čeksteryte V, Kurtinaitiene B, Navakauskiene R. 2018. Proteomic identification and enzymatic activity of buckwheat (*Fagopyrum esculentum*) honey based on different assays. *J. Food Nutr. Res.* **57**(1), 57-69. (ISSN 1336-867)
- Jasicka-Misiak I, Poliwođa A, Dereń M, Kafarski P. 2012. Phenolic compounds and abscisic acid as potential markers for the floral origin of two Polish unifloral honeys. *Food Chem.* **131**(4), 1149-1156. (doi: 10.1016/j.foodchem.2011.09.083)
- Kuś PM, Congiu F, Teper D, Sroka Z, Jerković I, Tuberoso CIG. 2014. Antioxidant activity, color characteristics, total phenol content and general HPLC fingerprints of six Polish unifloral honey types. *LWT- Food Sci. Technol.* **55**, 124-130. (doi: 10.1016/j.lwt.2013.09.016)
- Cheng N, Wang Y, Cao W. 2017. The Protective Effect of Whole Honey and Phenolic Extract on Oxidative DNA Damage in Mice Lymphocytes Using Comet Assay. *Plant Foods Hum. Nutr.* **72**, 388-395. (doi: 10.1007/s11130-017-0634-1)
- Kalinova J, Triska J, Vrchotova N. 2006. Distribution of vitamin E, squalene, epicatechin, and rutin in common buckwheat plants (*Fagopyrum esculentum* Moench). *J. Agric. Food Chem.* **54**(15), 5330-5335. (doi: 10.1021/jf060521r)
- Suzuki T, Morishita T, Kim SJ, Park SU, Woo SH, Noda T, Takigawa S. 2015. Physiological Roles of Rutin in the Buckwheat Plant. *Review. Jpn. Agr. Res. Q.* **49**(1), 37-43. (doi: 10.6090/jarq.49.37)
- Aleksenko SS. 2013. Antioxidant activity and phenolic compounds of buckwheat and barley by the data of spectrophotometry and HPLC. *J. Anal. Chem.* **68**, 458-465. (doi: 10.1134/S106193481305002X)
- Ölschläger C, Regos I, Zeller FJ, Treutter D. 2008. Identification of galloylated propylarionidins and procyanidins in buckwheat grain and quantification of rutin and flavanols from homostylous hybrids originating from *F. esculentum* x *F.*

- homotropicum*. *Phytochemistry* **69**, 1389–1397. (doi: 10.1016/j.phytochem.2008.01.001.)
14. Inglett GE, Chen D, Berhow M, Lee S. 2011. Antioxidant activity of commercial buckwheat flours and their free and bound phenolic compositions. *Food Chem.* **125**, 923–929. (doi: 10.1016/j.foodchem.2010.09.076)
15. Martín-García B, Pasini F, Verardo V, Gómez-Caravaca AM, Marconi E, Caboni MF. 2019. Antioxidant activity of commercial buckwheat flours and their free and bound phenolic compositions. *Foods* **8**(12), 670. (doi: 10.1016/j.foodchem.2010.09.076)
16. Wang KL, Zhang YJ, Yang CR. 2005. Antioxidant phenolic compounds from rhizomes of *Polygonum paleaceum*. *J. Ethnopharmacol.* **96**, 483–487. (doi: 10.1016/j.jep.2004.09.036)
17. Bittner K, Rzeppa S, Humpf HU. 2013. Distribution and Quantification of Flavan-3-ols and Procyanidins with Low Degree of Polymerization in Nuts, Cereals, and Legumes. *J. Agr. Food Chem.* **61**, 9148–9154. (doi: 10.1021/jf4024728)
18. Beckh G, Camps G. 2009. Neue Spezifikationen für Trachthonige. In Originalarbeiten, *Deut. Lebensm.-Rundsch.* **105**(2), 105–110. Available at: https://www.tentamus.com/qsi-de/wp-content/uploads/sites/11/2018/01/30_DLR-2009-105-2-Sortenspezifikationen.pdf
19. Popović V, Sikora V, Berenji J, Filipović V, Dolijanović Ž, Ikanović J, Dončić D. 2014. Analysis of buckwheat production in the world and Serbia. *Agric. Econ.* **61**(1), 53–62. (doi: 10.5937/ekoPolj1401053P)
20. Semikw P, Skowronek W, Teper D, Skubida P. 2008. Changes occurring in honey during ripening under controlled conditions based on pollen analysis and electrical conductivity. *J. Apic. Sci.* **52**(2), 45–53.
21. Panseri S, Manzo A, Chiesa LM, Giorgi A. 2013. Melissopalynological and volatile compounds analysis of buckwheat honey from different geographical origins and their role in botanical determination. *J. Chem.* **2013**, ID 904202. (doi: 10.1155/2013/904202)
22. Von Der Ohe W, Oddo L, Piana M, Morlot M, Martin P. 2004. Harmonized methods of melissopalynology. *Apidologie* **35**(Suppl. 1), S18–S25. (doi: 10.1051/apido:2004050)
23. Bucher E, Kofler V, Vorwohl G, Zieger E. 2004. Das Pollenbild der Südtiroler Honige. *Biologisches Labor der Landesagentur für Umwelt und Arbeitsschutz*, Bolzano, Italy, 17–37.
24. Riccardelli D'Albore G. 1997. Textbook of Melissopalynology. *Apimondia Publishing House*, Bucharest.
25. Vergeron P. 1964. Interprétation statistique des résultats en matière d'analyse pollinique des miels. *Les Annales de L'Abeille, INRA Editions* **7**(4), 349–364. (doi: 10.1051/apido:19640407)
26. Louveaux J, Maurizio A, Vorwohl G. 1978. Methods of Melissopalynology. *Bee World* **59**(4), 139–157. (doi: 10.1080/0005772X.1978.11097714)
27. Bogdanov, S. 2009. Harmonised methods of the international honey commission. *Bee Product Science*, International Honey Commission. Available at: <http://www.bee-hexagon.net/en/network.htm>
28. Gašić U, Šikoparija B, Tosti T, Trifković J, Milojković-Opsenica D, Natić M, Tešić Ž. 2014. Phytochemical Fingerprints of Lime Honey Collected in Serbia. *J. AOAC Int.* **97**(5), 1259–1267. (doi: 10.5740/jaoacint.SGEGasić)
29. Gašić U, Natić M, Mišić DM, Lušić DV, Milojković-Opsenica D, Tešić Ž, Lušić D. 2015. Chemical markers for the authentication of unifloral *Salvia officinalis* L. honey. *J. Food Compos. Anal.* **44**, 128–138.
30. Nešović M, Gašić U, Tosti T, Trifković J, Baošić R, Blagojević S, Ignjatović Lj, Tešić Ž. 2020. Physicochemical analysis and phenolic profile of polyfloral and honeydew honey from Montenegro. *RSC Adv.* **10**, 2462–2471. (doi: 10.1039/C9RA08783D)
31. Somerville A. 2001. Nutritional Value of Bee Collected Pollens. Rural Industries Research and Development Corporation. *NSW Agriculture*, Goulburn NSW. Available at http://www.nbba.ca/wp-content/uploads/2013/12/Nutritional_Value_of_Bee_Collected_Pollens.pdf.
32. Kaškonien V, Venskutonis PR. 2010. Floral Markers in Honey of Various Botanical and Geographic Origins: A Review. *Compr. Rev. Food Sci. Food Saf.* **10**, 620–634. (doi: 10.1111/j.1541-4337.2010.00130.x)
33. Bohm K, Thomazo L. 2016. Pollen Spectra of selected monofloral and polyfloral honeys of China. Presented at the Chinese Honey Workshop, Bologna, Italy.
34. Stanek N, Jasicka-Misiak I. 2018. HPTLC Phenolic Profiles as Useful Tools for the Authentication of Honey. *Food Anal. Methods* **11**, 2979–2989. (doi: 10.1007/s12161-018-1281-3)
35. Bryant VM Jr, Jones GD. 2001. The R-Values of Honey: Pollen Coefficients. *Palynology* **25**(1), 11–28. (doi: 10.1080/01916122.2001.9989554)
36. Cawoy V, Kinet JM, Jacquemart AL. 2008. Morphology of Nectaries and Biology of Nectar Production in the Distylous Species *Fagopyrum esculentum*. *Ann. Bot.* **102**(5), 675–684. (doi: 10.1093/aob/mcn150)
37. Garcia-Robledo C, Mora F. 2007. Pollination Biology and the Impact of Floral Display, Pollen Donors, and Distyly on Seed Production in *Arcytophyllum lavarum* (Rubiaceae). *Plant Biol.* **9**(4), 453–461. (doi: 10.1055/s-2007-964962)
38. Directive 2014/63/EU of the European Parliament and of the Council amending Council Directive 2001/110/EC relating to honey. 2014. *OJEC*, L 164/1.
39. Popek S, Halagarda M, Kurska K. 2017. A new model to identify botanical origin of Polish honeys based on the physicochemical parameters and chemometric analysis. *LWT-Food Sci. Technol.* **77**, 482–487. (doi: 10.1016/j.lwt.2016.12.003)
40. Bogdanov S, Jurendic T, Sieber R, Gallmann P. 2008. Honey for Nutrition and Health: a Review. *J. Am. Coll. Nutr.* **27**, 677–689. (doi: 10.1080/07315724.2008.10719745)
41. A. Salonen, V. Virjamo, P. Tammela, L. Fauch and R. Julkunen-Tiitto. 2017. Screening bioactivity and bioactive constituents of Nordic unifloral honeys. *Food Chem.*, **237**, 214–224. (doi: 10.1016/j.foodchem.2017.05.085)
42. A. Pavlović, A. Papetti, D. Dabić Zagorac, U. Gašić, D. Mišić, Ž. Tešić and M. Natić. 2016. Phenolics composition of leaf extracts of raspberry and blackberry cultivars grown in Serbia. *Ind. Crop. Prod.*, **87**, 304–314. (doi: 10.1016/j.indcrop.2016.04.052)
43. Truchado P, Ferreres F, Tomas-Barberan FA. 2009. Liquid chromatography-tandem mass spectrometry reveals the widespread occurrence of flavonoid glycosides in honey, and their potential as floral origin markers. *J. Chromatogr. A*, **1216**, 7241–7248.
44. Vasić V, Gašić U, Stanković D, Lušić D, Vukić-Lušić D, Milojković-Opsenica D, Tešić Ž, Trifković J. 2019. Towards better quality criteria of European honeydew honey: Phenolic profile and antioxidant capacity. *Food Chem.*, **274**, 629–641. (doi: 10.1016/j.foodchem.2018.09.045)
45. S. Kečkeš, U. Gašić, T. Čirković-Veličković, D. Milojković-Opsenica, M. Natić and Ž. Tešić. 2013. The determination of phenolic profiles of Serbian unifloral honeys using ultra-high-performance liquid chromatography/high resolution accurate mass spectrometry. *Food Chem.*, **138**, 32–40. (doi: 10.1016/j.foodchem.2012.10.025)
46. F. Cuyckens and M. Claeys, 2005. Determination of the glycosylation site in flavonoid mono-O-glycosides by collision-induced dissociation of electrospray-generated deprotonated and sodiated molecules. *J. Mass Spectrom.*, **40**, 364–372. (doi: 10.1002/jms.794)
47. Truchado P, Ferreres F, Bortolotti L, Sabatini AG, Tomás-Barberán FA. 2008. Nectar Flavonol Rhamnosides Are Floral Markers of Acacia (*Robinia pseudacacia*) Honey. *J. Agr. Food Chem.*, **56**, 8815–8824. (doi: 10.1021/jf801625t)
48. Truchado P, E. Tourn, L. M. Gallez, D. A. Moreno, F. Ferreres and F. A. Tomás-Barberán FA. 2010. Identification of Botanical Biomarkers in Argentinean Diplotaxis Honeys: Flavonoids and Glucosinolates. *J. Agr. Food Chem.*, **58**, 12678–12685. (doi:10.1021/jf103589c)
49. S. Kumar and K. A. Pandey, 2013. Chemistry and Biological Activities of Flavonoids: An Overview. *Sci. World J.*, **2013**, Article ID 162750. (doi: 10.1155/2013/162750)
50. D. Jang, Y. S. Jung, M.-S. Kim, S. E. Oh, T. G. Nam and D. O. Kim, 2019. Developing and Validating a Method for Separating Flavonoid Isomers in Common Buckwheat Sprouts Using HPLC-PDA. *Foods*, **8**, 549. (doi: 10.3390/foods8110549)
51. Tomás-Barberán FA, Ferreres F, Carcia-Viguera C, Tomás-Lorente F. 1993. Flavonoids in honey of different geographical origin. *Z. Lebensm.-Unters. –Forsch.*, **196**, 38–44. (doi: 10.1007/BF01192982)
52. R. Socha, L. Juszcak, S. Pietrzyk, D. Gałkowska, T. Fortuna and T. Witczak. 2011. Phenolic profile and antioxidant properties of Polish honeys. *Int. J. Food Sci. Techn.* **46**, 528–534. (doi: 10.1111/j.1365-2621.2010.02517.x)
53. Shen S, Wang J, Chen X, Liu T, Zhuo Q, Zhang SQ. 2019. Evaluation of cellular antioxidant components of honeys using UPLC-MS/MS and HPLC-FLD based on the quantitative composition-activity relationship.

Food Chem. **293**, 169-177. (doi:
 10.1016/j.foodchem.2019.04.105)

Tables

Table 1. List of investigated honey samples and melissopalynological analysis which declared samples to be buckwheat honey types after pollen from nectarless plants is excluded.

Sample	Location	% of Fagopyrum pollen	Secondary pollen (16-45 %)	Important minor pollen (3-15 %)
H1	Serbia	4.05	Echium , Astragalus type	Fagopyrum , Achillea type, Aster type, Filipendula
H2	Serbia	8.91	Astragalus type	Fagopyrum , Filipendula , Lamiaceae S type, Hypericum , Brassicaceae , Amorpha type,
H3	Serbia	10.93	Filipendula , Amorpha type	Fagopyrum , Astragalus type, Rubus , Teucrium , Clematis
H4	Serbia	4.47	Filipendula	Fagopyrum , Rosaceae (Prunus type), Rubus , Astragalus type, Trifolium pratense , Fenestrata , Amorpha type, Tilia , Robinia , Teucrium , Rhamnus type,
H5	Poland	40.83	Fagopyrum , Brassica napus	Trifolium pratense , Phacelia , Centaurea cyanus , Senecio type
H6	Poland	24.29	Fagopyrum , Rubus , Brassicaceae	Astragalus type, Fenestrata

Pollen frequency classes: P - "Predominant pollen" (more than 45 % of pollen grains counted), S - "Secondary pollen" (16-45 %); I - "Important minor pollen" (3-15 %); M - "Minor important pollen" (less than 3 %).

Table 2. Physicochemical parameters (water content, electrical conductivity, sugar content) of buckwheat honey samples from Serbia (H1-H4) and Poland (H5, H6).

Physicochemical parameters	Serbian honeys					Polish honeys		
	H1	H2	H3	H4	Mean ± SD	H5	H6	Mean ± SD
Moisture content (%)	17.10	17.03	15.55	15.50	16.29 ± 0.89	15.50	15.55	15.53 ± 0.04
Electrical conductivity (mS/cm)	0.41	0.43	0.35	0.32	0.38 ± 0.05	0.279	0.212	0.25 ± 0.05
Sugars (g/100 g)	67.00	69.73	68.46	67.39	68.15 ± 1.06	71.53	71.16	71.34 ± 0.26
Glucose (g/100 g)	25.91	28.47	25.12	24.57	26.02 ± 1.72	27.51	26.70	27.11 ± 0.57
Fructose (g/100 g)	37.39	37.26	39.03	38.61	38.07 ± 0.88	39.61	39.39	39.50 ± 0.16
Sucrose (g/100 g)	1.304	1.301	1.300	1.279	1.30 ± 0.01	1.286	1.789	1.54 ± 0.36
Maltose (g/100 g)	0.681	0.633	0.537	0.532	0.60 ± 0.07	0.676	0.741	0.71 ± 0.05
Isomaltose (g/100 g)	0.418	0.593	0.804	0.773	0.65 ± 0.18	0.695	0.606	0.65 ± 0.06
Trehalose (g/100 g)	0.123	0.151	0.186	0.182	0.16 ± 0.03	0.078	0.079	0.08 ± 0.00
Turanose (g/100 g)	0.721	0.908	0.998	0.948	0.89 ± 0.12	0.926	0.958	0.94 ± 0.02
Melibiose (g/100 g)	0.252	0.213	0.264	0.247	0.24 ± 0.02	0.353	0.359	0.36 ± 0.00
Melezitose (g/100 g)	0.209	0.207	0.221	0.252	0.22 ± 0.02	0.388	0.530	0.46 ± 0.10
Sum of monosaccharides (g/100 g)	63.29	65.72	64.16	63.18	64.09 ± 1.17	67.13	66.09	66.61 ± 0.73
Sum of disaccharides (g/100 g)	3.50	3.80	4.09	3.96	3.84 ± 0.25	4.01	4.53	4.27 ± 0.37

Mean ± SD - Mean value ± standard deviation (p ≤ 0.05).

Table 3. High-resolution MS data and negative ion MS², MS³ and MS⁴ fragmentation of phenolic compounds identified in Serbian buckwheat honey H1, Poland buckwheat honey sample H5, nectar (N) and pollen (P).

No	Compound name	t _R , min	Molecular formula, [M-H] ⁻	Calculated mass, [M-H] ⁻	Exact mass, [M-H] ⁻	Δ, ppm	MS ² Fragments, (% Base Peak)	MS ³ Fragments, (% Base Peak)	MS ⁴ Fragments, (% Base Peak)	H1	H5	N	P
Benzoic acid derivatives													
1	Gallic acid	2.39	C ₇ H ₅ O ₅ ⁻	169.014	169.014	0.2	84(3), 123(8), 124(5), 125(100) , 126(8)	69(49), 79(8), 81(93), 83(53), 97(100)	ND	-	-	-	+
2	Protocatechuic acid [†]	4.50	C ₇ H ₅ O ₄ ⁻	153.019	153.019	1.0	107(3), 109(100) , 110(8), 123(7)	65(42), 81(100) , 91(68), 106(17)	ND	+	+	+	+
3	p-Hydroxybenzoic acid [†]	5.50	C ₇ H ₅ O ₃ ⁻	137.024	137.024	1.2	93(100) , 94(6), 109(3)	ND	ND	+	+	+	+

Cinnamic acid derivatives														
1	4	3-O-Caffeoylquinic acid	4.72	C ₁₆ H ₁₇ O ₉	353.087	353.086	4.7	135(6), 179(29), 191(100), 192(4)	85(100), 93(62), 127(95), 173(73)	ND	+	+	-	-
2				-	81	14	3							
3	5	Caffeoyl hexoside	5.26	C ₁₅ H ₁₇ O ₉	341.087	341.087	0.1	135(10), 179(100), 180(9)	135(100)	107(100)	+	-	-	-
4				-	81	75	5							
5	6	5-O-Caffeoylquinic acid ¹	5.36	C ₁₆ H ₁₇ O ₉	353.087	353.087	0.0	179(3), 191(100)	85(100), 93(66), 127(89), 173(70)	57(100)	+	+	+	+
6				-	81	8	1							
7	7	Coumaroyl hexoside	5.39	C ₁₅ H ₁₇ O ₈	325.092	325.092	0.1	119(9), 163(100), 164(5), 289(18)	119(100)	ND	-	+	+	-
8				-	89	86	0.1							
9	8	Caffeic acid ¹	5.89	C ₉ H ₇ O ₄ ⁻	179.034	179.034	0.0	134(7), 135(100)	91(56), 107(100), 117(16)	ND	+	+	+	-
10				-	98	97	5							
11	9	5-O-p-Coumaroylquinic acid	6.40	C ₁₆ H ₁₇ O ₈	337.092	337.092	0.5	163(3), 191(100), 192(4)	85(100), 93(55), 127(95), 173(68)	ND	-	+	-	-
12				-	89	72	0.5							
13	10	Methyl 5-O-caffeoylquininate	6.51	C ₁₇ H ₁₉ O ₉	367.103	367.102	3.8	135(44), 161(11), 179(100), 191(20)	135(100)	79(53), 107(100), 151(18)	-	-	-	+
14				-	46	03	7							
15	11	p-Coumaric acid ¹	6.78	C ₉ H ₇ O ₃ ⁻	163.040	163.039	2.2	119(100), 120(4)	91(5), 93(100)	ND	+	+	+	-
16				-	07	7	7							
17	12	Cinnamic acid	7.01	C ₉ H ₇ O ₂ ⁻	147.045	147.044	4.1	103(100)	ND	ND	+	+	+	-
18				-	15	54	6							
19	13	Ferulic acid ¹	8.22	C ₁₀ H ₉ O ₄ ⁻	193.050	193.050	1.4	134(34), 147(100), 161(47), 178(15)	101(13), 103(22), 111(8), 129(100)	55(9), 57(60), 73(3), 85(100)	-	+	-	-
20				-	63	36	1							
21	14	Benzyl caffeate	11.5	C ₁₆ H ₁₃ O ₄	269.081	269.081	1.1	134(100), 135(5), 178(44), 225(8)	106(100), 108(16), 121(6), 150(29)	ND	+	+	+	-
22				-	93	63	3							
23	15	Prenyl caffeate	11.5	C ₁₄ H ₁₅ O ₄	247.097	247.097	0.6	135(17), 161(3), 179(100), 180(5)	135(100)	65(8), 79(7), 107(100), 117(5)	+	+	+	-
24				-	58	41	9							
25	16	Cinnamyl caffeate	12.5	C ₁₈ H ₁₅ O ₄	295.097	295.097	0.0	134(100), 178(89), 211(46), 251(49)	106(100), 109(12), 121(38)	ND	+	+	+	-
26				-	58	56	9							
Flavan-3-ol monomers and dimers														
27	17	B type procyanidin dimer gallate	5.17	C ₃₇ H ₂₉ O ₁₇	745.141	745.138	3.4	423(12), 441(38), 467(21), 593(100)	289(8), 423(17), 441(100), 467(30)	153(36), 287(17), 289(21), 315(100)	-	-	-	+
28				-	02	43	8							
29	18	Methyl-B type prodelfinidin dimer	5.61	C ₃₁ H ₂₇ O ₁₁	607.145	607.144	1.9	287(45), 405(47), 423(15), 455(100)	315(8), 405(100), 423(27), 437(82)	243(100), 283(4)	-	-	-	+
30				-	71	53	4							
31	19	Epicatechin	5.93	C ₁₅ H ₁₃ O ₆	289.071	289.070	3.5	179(9), 205(28), 245(100), 246(6)	161(19), 187(25), 188(13), 203(100)	161(33), 175(100), 185(21), 188(65)	-	-	-	+
32				-	76	74	3							
33	20	B type procyanidin dimer gallate isomer 1	6.18	C ₃₇ H ₂₉ O ₁₁	729.146	729.143	3.1	289(22), 407(100), 559(73), 577(63),	243(19), 255(21), 283(30), 285(100)	213(4), 241(4), 257(100)	-	-	-	+
34				-	11	78	9							
35	21	(Epi)catechin gallate	6.84	C ₂₂ H ₁₇ O ₁₁	441.082	441.081	1.6	169(15), 289(100), 303(3), 331(11)	179(12), 205(34), 231(6), 245(100)	161(19), 187(20), 188(13), 203(100)	-	-	-	+
36				-	72	98	7							
37	22	Methyl-(epi)gallocatechin gallate	7.03	C ₂₃ H ₁₉ O ₁₁	471.093	471.092	1.3	169(15), 287(100), 319(34), 439(42)	125(100), 161(9), 243(14), 245(3)	57(100)	-	-	-	+
38				-	29	66	4							
39	23	B type procyanidin dimer gallate isomer 2	7.19	C ₃₇ H ₂₉ O ₁₁	729.146	729.144	2.4	407(100), 441(22), 559(59), 577(46)	283(33), 285(100), 297(35), 389(19)	213(5), 241(3), 257(100)	-	-	-	+
40				-	11	35	2							
Flavones														
41	24	Luteolin 6-C-hexoside	6.15	C ₂₁ H ₁₉ O ₁₁	447.093	447.092	0.8	327(100), 328(11), 357(33), 358(6)	284(6), 299(100), 300(7)	175(48), 213(66), 255(100), 271(47)	-	+	-	+
42				-	29	91	4							
43	25	Luteolin 8-C-hexoside	6.31	C ₂₁ H ₁₉ O ₁₁	447.093	447.091	3.3	327(100), 357(35), 358(3), 369(5), 393(3)	191(3), 255(3), 284(17), 299(100)	175(41), 213(61), 240(43), 255(100)	-	+	-	-
44				-	29	77	8							
45	26	Apigenin 8-C-hexoside (Vitexin) ¹	6.65	C ₂₁ H ₁₉ O ₁₁	431.098	431.097	2.6	311(100), 312(11), 341(15), 342(3)	283(100), 284(3)	183(47), 211(29), 224(50)	-	+	-	+
46				-	37	23	4							

27	Luteolin ¹	8.79	C ₁₅ H ₉ O ₆ ⁻	285.040 46	285.039 7	2.6 8	151(39), 199(86), 241 (100), 243(60)	185(15), 197 (100), 199(80), 213(58)	152(16), 155(13), 169 (100), 179(15)	+	+	+	-
28	Apigenin ¹	9.65	C ₁₅ H ₉ O ₅ ⁻	269.045 55	269.044 8	2.7 9	149(8), 151(5), 201(6), 225 (100), 226(15)	181(13), 183(5), 197(3), 207 (100)	ND	+	+	+	-
29	Chrysoeriol	10.1 4	C ₁₆ H ₁₁ O ₆ -	299.056 11	299.056 09	0.0 7	284 (100), 285(13)	227(8), 255 (100), 256(10)	187(8), 211(92), 213(10), 227 (100)	+	+	+	-
30	Genkwanin ¹	10.5 9	C ₁₆ H ₁₁ O ₅ -	283.061 2	283.061 11	0.3 2	211(17), 239(82), 240(31), 268 (100)	211(17), 239 (100), 240(30)	195(73), 211 (100), 239(4)	+	-	-	-
31	Tricin	11.2 2	C ₁₇ H ₁₃ O ₇ -	329.066 68	329.066 59	0.2 8	314 (100), 315(11)	271(7), 285(3), 299 (100)	243(5), 255(4), 271 (100)	+	+	+	-
32	Chrysin ¹	11.7 5	C ₁₅ H ₉ O ₄ ⁻	253.050 63	253.050 56	0.3	207(34), 209 (100), 210(13), 211(16)	165(53), 167(16), 180(88), 181 (100)	139 (100), 152(10), 153(99), 156(6)	+	+	+	-
33	Acacetin ¹	12.3 9	C ₁₆ H ₁₁ O ₅ -	283.061 2	283.061 16	0.1 1	268 (100), 269(10)	211(11), 239 (100), 240(18)	195(62), 211 (100), 239(3)	+	+	+	-
Flavonols													
34	Myricetin	6.23	C ₁₅ H ₉ O ₈ ⁻	317.030 29	317.030 15	0.4 3	163(14), 191 (100), 207(23), 299(31)	135(5), 163 (100)	91(4), 107(24), 119(56), 135 (100)	+	-	-	+
35	Quercetin 3-O-(6"-rhamnosyl)-hexoside (Rutin) ¹	6.51	C ₂₇ H ₂₉ O ₁ 6 ⁻	609.146 11	609.143 85	3.7 1	179(3), 300(31), 301 (100), 343(6)	151(75), 179 (100), 271(47), 273(19)	151 (100)	+	+	+	+
36	Quercetin 3-O-galactoside ¹	6.77	C ₂₁ H ₁₉ O ₁ 2 ⁻	463.088 2	463.087 74	1	300(36), 301 (100), 302(13)	151(79), 179 (100), 255(25), 271(34)	151 (100)	+	+	-	+
37	Quercetin 3-O-rhamnoside ¹	7.27	C ₂₁ H ₁₉ O ₁ 1 ⁻	447.093 29	447.092 61	1.5 1	300(21), 301 (100), 302(8)	151(83), 179 (100), 255(27), 271(34)	151 (100)	+	+	+	+
38	Kaempferol 3-O-rhamnoside	7.78	C ₂₁ H ₁₉ O ₁ 0 ⁻	431.098 37	431.097 87	1.1 7	255(6), 284(49), 285 (100), 286(15)	229(30), 255 (100), 256(62), 257(80)	210(5), 211(59), 212(4), 227 (100)	+	-	+	+
39	Quercetin ¹	8.85	C ₁₅ H ₉ O ₇ ⁻	301.035 38	301.035 16	0.7 2	151(82), 179 (100), 257(12), 273(15)	151 (100)	63(4), 65(3), 83(13), 107 (100)	+	+	+	+
40	Quercetin 3-methyl ether	9.16	C ₁₆ H ₁₁ O ₇ -	315.051 03	315.050 66	1.1 6	300 (100), 301(10)	254(9), 255(52), 271 (100), 272(6)	199(16), 215(18), 227(67), 243 (100)	+	+	+	-
41	Kaempferol ¹	9.82	C ₁₅ H ₉ O ₆ ⁻	285.040 46	285.040 22	0.8 5	151(71), 185(83), 229 (100), 239(78)	187(41), 201(93), 211(43)	157 (100), 167(16), 137(24)	+	+	+	+
42	Herbacetin 8-methyl ether	9.82	C ₁₆ H ₁₁ O ₇ -	315.051 03	315.050 89	0.4 3	300 (100), 301(10)	255(59), 256(56), 271(23), 272 (100)	166(51), 216(30), 244 (100)	+	+	+	-
43	Isorhamnetin	10.0 1	C ₁₆ H ₁₁ O ₇ -	315.051 03	315.050 92	0.3 3	300 (100), 301(11)	151 (100), 227(49), 271(96), 272(76)	83(6), 107 (100)	+	+	+	-
44	Dimethyl quercetin	10.3 2	C ₁₇ H ₁₃ O ₇ -	329.066 68	329.066 5	0.5 2	314 (100), 315(11)	271(3), 299 (100)	227(6), 243(5), 255(9), 271 (100)	+	+	+	-
45	Rhamnetin	10.8 5	C ₁₆ H ₁₁ O ₇ -	315.051 03	315.050 81	0.6 9	165 (100), 193(33), 287(15), 300(44)	65(24), 91(15), 97(53), 121 (100), 150(44)	89(28), 91 (100), 93(20), 106(17)	+	+	-	-
46	Galangin ¹	11.9 7	C ₁₅ H ₉ O ₅ ⁻	269.045 55	269.044 86	2.5 5	197(91), 198(50), 213 (100), 227(89)	141(20), 169(95), 185 (100), 195(36)	141 (100), 143(54), 157(31), 158(4)	+	+	+	-
47	Kaempferide ¹	12.0 3	C ₁₆ H ₁₁ O ₆ -	299.056 11	299.055 97	0.4 7	165(6), 271(5), 284 (100), 285(10)	151 (100), 164(24), 228(22), 255(20)	63(3), 65(3), 83(12), 107 (100), 122(5)	+	+	+	-

Flavanonols											
1											
2	48	Aromodedrin	8.01	C ₁₅ O ₁₁ O ₆	287.056	287.056	0.2	243(13),	125(50), 151(18),	158(12),	
3				-	11	03	7	259(100),	173(31), 215(100)	172(30),	+ + + -
4								260(12), 269(5)		173(100),	
5										200(16)	
6	49	Pinobanksin 5-methyl ether	9.16	C ₁₆ H ₁₃ O ₅	285.076	285.076	0.6	239(23), 252(16),	223(5), 224(8),	180(12),	
7				-	85	66	6	267(100), 268(10)	239(3), 252(100)	208(70),	+ + + -
8										223(25),	
9	50	Pinobanksin	9.94	C ₁₅ H ₁₁ O ₅	271.061	271.060	2.0	151(10), 197(18),	165(14), 181(27),	224(100)	
10				-	2	64	5	225(28), 253(100)	209(61), 225(100)	181(27), 183(6),	+ + + -
11										197(100)	
12	51	Pinobanksin 3-acetate	12.0	C ₁₇ H ₁₃ O ₆	313.071	313.071	0.2	253(100),	165(12), 181(16),	153(24),	
13				-	76	69	4	254(11), 271(13)	209(100), 211(13)	165(80),	+ + + -
14										180(96),	
15										181(100)	
16	52	Pinobanksin 3-propanoate	12.9	C ₁₈ H ₁₅ O ₆	327.087	327.087	1.1	253(100),	165(14), 181(17),	165(48),	
17				-	41	05	2	254(11), 271(5)	209(100), 211(11)	167(14),	+ - - -
18										180(47),	
19										181(100)	
20	53	Pinobanksin 3-butyrate	13.7	C ₁₉ H ₁₇ O ₆	341.103	341.102	1.2	253(100), 254(11)	165(10), 181(18),	165(32),	
21				-	06	63	8		209(100), 211(15)	167(13),	+ + - -
22										180(26),	
23										181(100)	
24	54	Pinobanksin 3-pentanoate	14.5	C ₂₀ H ₁₉ O ₆	355.118	355.118	0.9	253(100),	165(11), 181(18),	153(25),	
25				-	71	38	4	254(10), 291(11), 309(4)	185(10), 209(100)	165(58),	+ + + -
26										180(61),	
27										181(100)	
28	Flavanone										
29	55	Pinocembrin ¹	11.8	C ₁₅ H ₁₁ O ₄	255.066	255.065	4.5	151(32), 187(14),	145(18), 169(23),	115(19),	
30				-	28	12	5	211(34), 213(100)	184(3), 185(100)	141(87),	+ + + -
31										143(100),	
32										157(18)	

¹Confirmed using available standards; ND – Not Detected.

Appendix B

Editorial office

October, 2020

Please find the new version of the article entitled: “**Polyphenolic profile of buckwheat honey, nectar and pollen**” (by Milica Nešović, Uroš Gašić, Tomislav Tosti, Nikola Horvacki, Branko Šikoparija, Nebojša Nedić, Stevan Blagojević, Ljubiša Ignjatović and Živoslav Tešić) which we would like to publish in your journal. Please note that the title of the paper has been changed due to the reviewer's request. We have corrected the paper according to the Reviewers comments and suggestions. The authors are grateful and find all the suggestions very useful. All changes, improvements and answers to comments are given in the text below and marked it was marked in **red**.

In the hope that our article will satisfy the criteria and research quality for publication in *Royal Society Open Science*, I remain, with personal regards,

Sincerely,

Živoslav Tešić, PhD

Full professor

University of Belgrade –Faculty of Chemistry

Belgrade, Serbia

Reviewers' Comments to Author:

Reviewer: 1

Comments to the Author(s)

The manuscript reports on the polyphenolic profile of buckwheat honey, nectar and pollen. The authors used state-of-the-art analytical methods to study the similarity of the polyphenolic profile of honey with the ones of nectar and pollen. These results provide important information for the scientific field. However, there are some issues that need to be addressed in order to improve the manuscript.

1. The manuscript requires an improvement in terms of language (grammatical, linguistic and spelling).

Answer: Thank you. We have improved the language of the Manuscript.

2. Regarding the number of samples analysed, not taking into account the diversity due to i.e. year of harvest, conclusions need to be formulated with more caution. With this in mind, I suggest revision and reformulation of:

- the last two sentences of the Summary;
- the Conclusions.

Answer: We have changed the sentence in the Summary and the Conclusion sections.

3. The term "different levels of monoflorality" is unusual and vague. Basing on the criteria set, a sample of honey is declared as monofloral or polyfloral, respectively. It holds true that for certain types of honey the criteria needs to be extended to declare monoflorality. I suggest to revise and rephrase this term throughout the manuscript.

Answer: Percentage of pollen indicates contribution of nectar of given pollen source (Bryant and Jones, 2001, ref [35]). Since the relation is not linear % of pollen is kept as a measure whether analyzed honey could be considered monofloral. We have proposed term "monoflorality" to refer to variability in *Fagopyrum* nectar contribution, which is indicated with observed variability in % of *Fagopyrum* pollen. We have adapted the following statements to avoid using term "monoflorality":

4. In Chapter 3.1 (Samples), please provide a short note on how the sample of nectar and pollen was collected and stored.

Answer: We added some details of the samples collection as well as the map of collection areas (see Figure 1).

5. In the Conclusions, please reformulate the part: "However, in the case of discrepancy, ..." bearing in mind that melissopalynological, physico-chemical and sensory analysis is used to determine / confirm botanical origin (Directive on honey, 2001) and the fact that buckwheat honey has characteristic sensory properties that allow it to be distinguished from other types of honey.

Answer: We have modified the sentence.

6. I suggest to omit word "significant" when describing differences, since statistical analysis was not performed as well as it is not in a place with this number of samples.

Answer: Thank you. We changed the word "significant" where is not appropriate.

7. In the supplementary material, in table S2, please check the sample labels for samples from Poland - there are two H5 labels and H6 is missing.

Answer: Thank you. It is changed now.

Reviewer: 2

Comments to the Author(s)

Dear authors,

This is an interesting manuscript. However I pointed some suggestions/modification in your manuscript.

I believe that can improve your work. I am attached the PDF file with my observations.

1. Page 3, Line 46: It will be interesting to have a map showing these two collection area.

Answer: We have made a map of areas (see Figure 1).

2. Page 3, Line 48: Which bee specie??

Answer: We added bee species *Apis mellifera*.

3. Page 3, Line 51: How was this sample collection? Could you explain more? How did you collect pollen and nectar from the flowers? Which period (year, flowering...)?

Answer: We added some details of the samples collection.

4. Page 4, Line 16: Are you sure about this terminology? For me, you could use "other elements was also counted and identified"....Its confuse this terminology because honeydew is a excretion of sugary liquids from homopterans that live as plant phloem-sucking parasites. Within this "honeydew", the elements mentioned in the text can be found. After all, are you working with honey or honeydew?

Answer: Thank you. We changed the term and modified this section.

5. Page 5, Line 9: I would like to see a plate with the principal identified pollen types in the analysed samples

Answer: Micrographs of pollen types contributing pollen spectrum with >15% were prepared and given in the supplementary as Figure S1.

6. Page 5, Line 12: **Corrected.**

7. Page 5, Line 14: **Corrected.**

8. Page 5, Line 39: **Corrected.**

9. Page 5, Line 52: **Corrected.**